# CONTEXTUALIZED SCENE IMAGINATION FOR GENERATIVE COMMONSENSE REASONING

**Peifeng Wang**[1,3], **Jonathan Zamora**[2*], **Junfeng Liu**[1*],
**Filip Ilievski**[3], **Muhao Chen**[1,3], **Xiang Ren**[1,3]
[1]Department of Computer Science, University of Southern California
[2]Department of Computer Science, University of California, San Diego
[3]Information Sciences Institute, University of Southern California
{peifengw,liujunfe,muhaoche,xiangren}@usc.edu,
jzamoraa@ucsd.edu, ilievski@isi.edu

## ABSTRACT

Humans use natural language to compose common concepts from their environment into plausible, day-to-day scene descriptions. However, such generative commonsense reasoning (GCSR) skills are lacking in state-of-the-art text generation methods. Descriptive sentences about arbitrary concepts generated by neural text generation models (e.g., pre-trained text-to-text Transformers) are often grammatically fluent but may not correspond to human common sense, largely due to their lack of mechanisms to capture concept relations, to identify implicit concepts, and to perform generalizable reasoning about unseen concept compositions. In this paper, we propose an Imagine-and-Verbalize (I&V) method, which learns to imagine a relational scene knowledge graph (SKG) with relations between the input concepts, and leverage the SKG as a constraint when generating a plausible scene description. We collect and harmonize a set of knowledge resources from different domains and modalities, providing a rich auxiliary supervision signal for I&V. The experiments demonstrate the effectiveness of I&V in improving language models on both concept-to-sentence and concept-to-story generation tasks, while enabling the model to learn well from fewer task examples and generate SKGs that make common sense to human annotators [1].

## 1 INTRODUCTION

Humans describe everyday scenes in natural language based on their understanding of common concepts encountered in their environment (Tincoff & Jusczyk, 1999). Analogously, the task of *generative commonsense reasoning* (GCSR) asks machines to generate a description of everyday situations based on a set of concepts and an initial context (Liu et al., 2020; Li et al., 2021). For example, given concept words {*dog, frisbee, catch, throw*}, a machine is expected to generate a plausible description, e.g., "*A man throws a frisbee and his dog catches it in the air*". Machines with GCSR skills would communicate fluidly with humans, e.g., when summarizing a document by preserving its key details (Sha, 2020), composing a creative story according to a set of clues (Yao et al., 2019), and generating a conversation reply that includes specified keywords (Mou et al., 2016).

GCSR poses three unique challenges for automatic text generation methods. To depict plausible scenes when composing sentences, machines require commonsense knowledge to reason about the relations between concepts and the affordances of objects (e.g., "*dog*" performs the action "*catch*" but not the action "*throw*"). Moreover, machines require a compositional generalization ability (Keysers et al., 2019), i.e., the ability to judge the plausibility of a new concept composition that has not been observed during training, and to identify concepts related to the scene that are not explicitly provided (e.g., "*person*" to perform "*throw*" in the above example).

GCSR can be directly attempted by fine-tuning pre-trained text-to-text language models (LMs) (Raffel et al., 2019; Radford et al., 2019). While pre-trained LMs capture certain encyclopedic knowl-

---

*Equal contributions

[1]Code and data are available at https://github.com/wangpf3/imagine-and-verbalize.

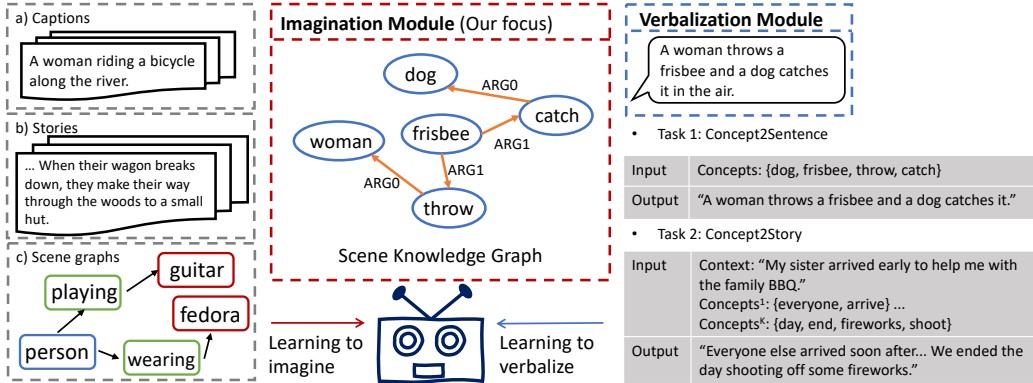

Figure 1: Overview of the proposed I&V method: (1) We leverage SKGs for unifying scene knowledge from different resources. (2) We pre-train a contextualized imagination module to construct an SKG for a set of concepts, based on the collected SKG instances. (3) At inference time, our verbalization module realizes the generated SKG into natural language.

edge mentioned in text corpora (e.g., Wikipedia) (Petroni et al., 2019) and can combine concepts in novel ways, they may generate grammatically fluent but implausible sentences that conflict with human common sense (Lin et al., 2020). This is because LMs have no intrinsic mechanism to reason over high-level relations between concepts Zhou et al. (2020). To close the knowledge gap, recent work augment LM input with knowledge graph triples (e.g., (*dog, CapableOf, catch*)) retrieved from ConceptNet (Liu et al., 2020; Li et al., 2020), or prototype sentences that cover input concepts retrieved from external text corpora (Fan et al., 2020; Wang et al., 2021). However, despite the input augmentation, GCSR skills are implicitly learned based on the concept-text pairs in the training data, without explicit supervision. While some recent work propose content planning in story generation in the form of plots or scripts (Yao et al., 2019; Fan et al., 2019), only the narrative order of concepts are planned in those methods instead of their plausible roles and relations. Given the complexity of the GCSR task, machines need a direct mechanism to create a high-level relational representation of the provided concepts, which would allow them to judge the plausibility of their combination.

In this paper, we propose to model an explicit *scene imagination* step which constructs a structured representation of a plausible scene based on input concepts and initial context. The scene imagination module formalizes the background knowledge required for the reasoning through a contextualized relational graph, called *scene knowledge graph* (SKG). An SKG allows us to collect and harmonize diverse commonsense knowledge across resources and modalities into a comprehensive SKG distribution (see Figure 1 for an illustration). We develop an imagine-and-verbalize framework: an imagination module learns to construct a contextualized SKG from input concepts and context by pretraining over a large amount of external SKGs; a verbalization module learns to faithfully realize the imagined SKG into natural language by training over downstream datasets. By learning from a large number of diverse SKGs, our method is able to capture plausible relations between concepts. By integrating these SKGs with LMs, the imagination module is able to compose new objects in novel ways, and identify implicit concepts for a scene. Imagine-and-verbalize decomposes the challenging scene description task into two realistic tasks for which a wealth of training data can be collected, simultaneously enabling for effective and explainable GCSR.

We experiment with two GCSR tasks and three scene graph resources, observing consistently better or competitive performance to SotA baselines. We find that (1) SKGs extracted from visual captions and story datasets are more helpful than other resources; (2) our model can learn faster (with less training data) with the help of scene imagination; and (3) the imagination module with a larger backbone LM demonstrates larger capacity in encoding commonsense knowledge. Our human evaluation study on the generated imagination indicates that these SKGs capture common sense and that the verbalization module generates the text by following the guidance of the imagination.

## 2 METHOD

Formally, in GCSR, we consider a list of *concepts sets* $\{\mathbf{x}^1, \mathbf{x}^2, ..., \mathbf{x}^K\}$ and a textual *context* $\mathbf{c} \in \mathcal{C}$ as input. Each concept set $\mathbf{x}^i$ is unordered and consists of multiple concept words $\{x_j\}$. A concept

word $x_j \in \mathcal{X}$ (or *concept* for brevity) is a commonly seen object (nouns such as "*dog*" or "*frisbee*") or commonly performed action (verbs such as "*throw*" or "*catch*"). The goal of GCSR is to generate $K$ sentences $\{\mathbf{y}^1, \mathbf{y}^2, ..., \mathbf{y}^K\}$, each describing a plausible situation following human common sense for a concept set $\mathbf{x}^i$. The $i$-th sentence $\mathbf{y}^i \subset \mathcal{Y}$ should be generated using all concepts in $\mathbf{x}^i$.

We consider two variants of GCSR: 1) *concepts-to-sentence* generation (Lin et al., 2020), where no context is given (i.e., $\mathbf{c}$ is empty) and only one concept set is provided ($K = 1$); and 2) *concepts-to-story* generation task, where $\mathbf{c}$ is the leading sentence of a multi-sentence story and more than one concept sets are provided, each corresponding to one sentence to be generated ($K > 1$). Both tasks are evaluated by comparing the machine-generated text with human-generated (gold) references.

## 2.1 THE IMAGINE-AND-VERBALIZE APPROACH

Pre-trained LMs struggle with learning a generalizable mapping from concepts to plausible sentences solely based on the training data. Augmenting concepts with external knowledge to form the input $\mathcal{X}'$ and fine-tuning a pretrained LM to model $P(\mathcal{Y}|\mathcal{C}, \mathcal{X}')$ (Liu et al., 2020; Fan et al., 2020; Li et al., 2021) alleviates this issue partially, while still learning a direct mapping of $\{\mathcal{C}, \mathcal{X}'\} \rightarrow \mathcal{Y}$. In this work (Figure 1), we decompose the GCSR task into two sub-tasks, namely contextualized scene generation (*imagination*) and scene-aware text generation (*verbalization*):

$$P(\mathcal{Y}|\mathcal{C}, \mathcal{X}) = \sum_{\mathcal{Z}} P(\mathcal{Y}|\mathcal{C}, \mathcal{X}, \mathcal{Z})P(\mathcal{Z}|\mathcal{C}, \mathcal{X}), \tag{1}$$

where $\mathcal{Z}$ denotes the scene representation for the given concepts and context.

The contextualized scene imagination module $P(\mathcal{Z}|\mathcal{C}, \mathcal{X})$ aims to construct a multi-relational graph representation $\mathcal{Z}$ (scene knowledge graph, or SKG) that describes a plausible scene that involves all input concepts and corresponds to the provided context. To learn this module, we collect a diverse set of SKG instances from different resources and modalities to form a comprehensive distribution of scenes (§2.2). The imagination module is pre-trained over the collected scene instances and learns to generate SKGs depicting plausible day-to-day situation. The imagination module is based on a neural architecture, which enables it to generate concept compositions that might not have been observed during training (§2.3).[2] We leverage the contextualized SKG for text generation with a verbalization module $P(\mathcal{Y}|\mathcal{C}, \mathcal{X}, \mathcal{Z})$ which takes the context, concepts, and the generated SKG as input, and composes a grammatical and plausible scene description in natural language (§2.4).

To perform GCSR, where one or multiple concept sets are given, we apply the imagination module to sample $\mathbf{z}^i$. Since the marginalization over $\mathcal{Z}$ is generally intractable due to the complex structure of the SKGs, we only sample the most probable scene representation $\mathbf{z}^{*i}$ that maximizes $P(\mathbf{z}^i|\mathbf{c}', \mathbf{x}^i)$, where $\mathbf{c}'$ includes the given context $\mathbf{c}$ and the previously generated $\mathbf{y}^j, (j < i)$ . We then apply the verbalization module to generate one sentence at a time by sampling from $P(\mathbf{y}^i|\mathbf{c}', \mathbf{x}^i, \mathbf{z}^{i*})$. Multiple sentences are generated by iteratively applying the imagination and verbalization modules.

## 2.2 IMAGINATION VIA GENERATING SKG

**Imagination through SKGs** We adopt the term "scene graph" from the computer vision community, and we generalize it to a novel relational schema that represents knowledge from multiple modalities. Our SKG is defined as a relational graph $\mathcal{G} = (\mathcal{E}, \mathcal{R})$ that organizes a set of concepts in a coherent scene. The node set $\mathcal{E}$ of the graph includes both given and implicit concepts, while each relation (edge type) $r \in \mathcal{R}$ denotes how two concepts should be related. We follow the Abstract Meaning Representation (AMR) (Banarescu et al., 2013) schema to consider the core relations between two concepts, which corresponds to the commonsense knowledge required by GCSR. Table 7 in the appendix illustrates a few representative relations and their examples.

**Collecting Diverse SKGs** We consider two complementary modalities, text and vision, as some concepts and relationships are more likely to occur in one modality versus another. (1) *Textual Modality:* According to pragmatic principles of human language, people generally leave out expected details about common scenes (Grice, 1975). For this reason, we extract SKGs from visual captions and narrative stories, in which human annotators are asked to explicitly describe scenes that

---

[2]The imagination module can be further fine-tuned over the downstream datasets.

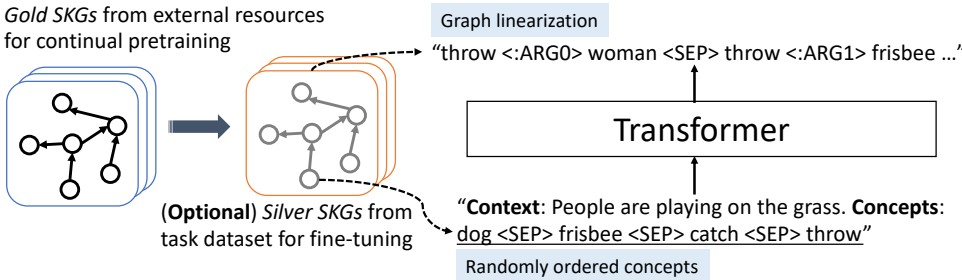

Figure 2: Continual pretraining and fine-tuning of the imagination module to output a linearized SKG based on a sequential input (context and concepts).

may happen using descriptive language as shown in Figure 1(a,b). To extract an SKG out of these textual signals, we adopt the AMR parsing tool to transform each sentence into an AMR graph. This process yields a single SKG per sentence. For the story SKGs, we also keep the sentences (up to 256 tokens) that precede the sentence that corresponds to the SKG, as context $\mathbf{c}$. (2) *Visual Modality:* Image captions focus on salient information and may not capture all useful visual signals. Thus, we also capture the scene structures directly from images, by using VisualGenome (Krishna et al., 2016), a large-scale scene graph dataset annotated by humans. To adopt a unified SKG schema, we manually map the relations in scene graphs from VisualGenome to the ones used in textual SKGs. A full set of mapping rules can be found in the Appendix (A.1). The statistics of the SKGs collected from each resource/modality are summarized in Table 1. We note that visual scene graphs may be biased towards knowledge about spatial relationships and object affordance, which further motivates our decision to extract SKGs from multiple modalities.

## 2.3   LEARNING THE SCENE IMAGINATION MODULE

We describe how we pre-train the scene imagination model using multimodal SKG examples collected from diverse sources, and how we fine-tune the imagination module to downstream datasets.

A straightforward way to construct a SKG is to retrieve instances that contains all the given concepts from the collected SKGs. However, performance of such method is limited by the coverage of the SKG collection and will fail when encountering novel concept composition. We

Table 1: Statistics of the SKG instances collected from different resources.

| Knowledge source | # SKGs | # Concepts |
|---|---|---|
| Caption-AMR | 584,252 | 22,961 |
| Story-AMR | 927,163 | 41,272 |
| VG-SceneGraph | 292,596 | 41,629 |
| All | 1,792,941 | 84,835 |

propose to model $P(\mathcal{Z}|\mathcal{C}, \mathcal{X})$ with a neural graph generator. Inspired by previous work on (conditional) graph generation (You et al., 2018), we formulate SKG construction as an auto-regressive sequence generation task, where a linearized SKG is generated sequentially conditioned on the context, input concepts, and the graph sequence generated so far. Since the sequence generation problem can be efficiently tackled by pre-trained auto-regressive LMs (e.g., GPT-2 Radford et al. (2019)), we adopt LMs as the backbone of our imagination module (Bosselut et al., 2019; Wang et al., 2020).

**Linearized SKG Generation**   To form training instances for the imagination module, we treat the nodes in an SKG instance as input concepts and the linearized SKG as the target output (Figure 2). The input concepts are concatenated into a sequence $\mathbf{x} = [x_1, x_2, ..., x_n]$, preceded by the context $\mathbf{c}' \in \mathcal{C}$. When $\mathbf{c}'$ is not given, we prepend the word "*none*" to the concept sequence. To linearize an AMR-based SKG into a sequence $\mathbf{z} = [z_1, z_2, ..., z_m]$, we adopt the PENMAN serialization format (Goodman, 2020) which converts AMR into a spanning tree over the graph. This format is shown to be more suitable than other linearization strategies like depth-first-search (DFS) in enabling LMs to learn the graph structure (Mager et al., 2020).

During training, we randomize the order of the concepts at every training step such that the graph generator learns to be invariant to concept order (Zhang et al., 2019). For each training instance, we randomly discard a small subset of the SKG nodes (concepts) in each training epoch. This simulates the scenario in which a subset of the concepts that constitute a scene will be given, thus teaching the model to infer implicit concepts for completing a plausible scene.

**Continual-Pretraining and Fine-tuning** With both the input concepts (plus context) and the output graph linearized as sequences based on the collected SKG instances, we continually pretrain an auto-regressive LM to generate $\mathbf{z} = \text{Transformer}(\mathbf{c}', \mathbf{x})$. The training objective is to maximize $P(\mathcal{Z}|\mathcal{C}, \mathcal{X})$ by minimizing the negative log-likelihood:

$$\mathcal{L}_{imagine} = -\sum_{t=1}^{t=m} \log P(z_t|z_{<t}, \mathbf{c}', \mathbf{x}). \quad (2)$$

Our pre-trained imagination module generates an SKG on the fly, and it can be further fine-tuned on downstream datasets, when their distributions of context and concepts are different from the pretraining data (see Figure 2 for illustration). Since downstream datasets cannot be expected to have ground-truth SKGs paired with each training example, we apply the AMR parsing tool described in §2.2 on the training sentences to obtain silver-standard SKGs. We then follow the same training procedure to continually pretrain the module into a customized imagination module for a specific downstream dataset.

## 2.4 SCENE-AWARE VERBALIZATION

**Iterative Imagine-and-Verbalize** At the inference time, we apply the trained imagination module iteratively to generate the most plausible SKG for each given concept set $\mathbf{x}^i$, i.e., $\mathbf{z}^{i*} = \arg\max_{\mathbf{z}^i} P(\mathbf{z}^i|\mathbf{c}', \mathbf{x}^i)$, where the context $\mathbf{c}'$ includes both the given context $\mathbf{c}$ and the previously generated sentences $\{\mathbf{y}^j\}$ ($j < i$). The generated SKG is used by the scene-aware verbalization module to model $P(\mathcal{Y}|\mathcal{C}, \mathcal{X}, \mathcal{Z})$. The verbalization module generates the $i$-th sentence by sampling from $P(\mathbf{y}^i|\mathbf{c}', \mathbf{x}^i, \mathbf{z}^{i*})$. Multiple sentences are generated iteratively by alternating between the scene imagination (to construct SKG) and verbalization (to produce the next sentence). See Figure 3 for an illustration of this iterative inference process.

**Model Training** Since both the linearized SKG (generated by the imagination module) and the target sentences are sequences by nature, we design $P(\mathcal{Y}|\mathcal{C}, \mathcal{X}, \mathcal{Z})$ as a sequence-to-sequence generative model and learn this verbalization module by fine-tuning another pre-trained auto-regressive LM, i.e., $\mathbf{y}^i = \text{Transformer}(\mathbf{c}', \mathbf{x}^i, \mathbf{z}^i)$. To form the input for generating the sentence $\mathbf{y}^i$, we concatenate the context $\mathbf{c}'$, the concept set sequence $\mathbf{x}^i$ and $\mathbf{z}^i$ into one sequence[3] as illustrated in Figure 3. We then train the model to maximize $P(\mathcal{Y}|\mathcal{C}, \mathcal{X}, \mathcal{Z})$ by minimizing the negative log-likelihood:

$$\mathcal{L}_{verbalize} = -\sum_{t=1}^{t=l} \log P(y_t^i|y_{<t}^i, \mathbf{c}', \mathbf{x}^i, \mathbf{z}^i). \quad (3)$$

For each training instance $(\mathbf{y}^i, \mathbf{c}', \mathbf{x}^i)$, we construct two types of SKG instances as the input $\mathbf{z}^i$:

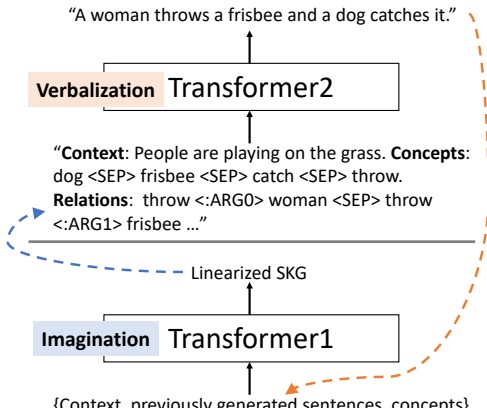

Figure 3: Our I&V method iteratively applies the imagination and the verbalization modules, by generating one sentence in each iteration.

(1) We perform AMR parsing on $\mathbf{y}^i$ to obtain a silver-standard SKG; (2) We apply the trained imagination module to generate a SKG $\mathbf{z}^{i*} = \arg\max_{\mathbf{z}^i} P(\mathbf{z}^i|\mathbf{c}', \mathbf{x}^i)$, where $\mathbf{c}'$ includes the given context $\mathbf{c}$ and the ground-truth prefix sentences $\{\mathbf{y}^j\}$ ($j < i$). We find it beneficial to train the verbalization module over these two types of SKGs as evidenced by our ablation study (§A.7). During inference, the SKG $\mathbf{z}^i$ is generated by the imagination module, while $\mathbf{c}'$ includes the given context $\mathbf{c}$ and the previous sentences $\{\mathbf{y}^j\}$ ($j < i$) generated by the verbalization module.

## 3 EXPERIMENTAL SETUP

**Tasks & Datasets** We consider two GCSR tasks: Concept2Sentence and Concept2Story. *(1) Concept2Sentence* is a task of generating a single sentence for a given set of concepts and no context.

---

[3]Our ablation study in Appendix A.6 shows that including all these elements as input is helpful.

We evaluate Concept2Sentence on the CommonGen (Lin et al., 2020) benchmark. Since the labels of the official test set are not publicly available, we submit our method to the leaderboard to obtain its performance. Notably, the concept sets in CommonGen's test set are novel and do not appear in the training set. We also create an in-house split of CommonGen to facilitate comparison between different variants of our method and the baselines. *(2) Concept2Story* is a generalization of the concept2sentence task, where the goal is to generate a coherent story with $K = 4$ sentences given a set of concepts and an initial verbal context. We construct two benchmarks based on the Visual Story Telling (VIST) (Huang et al., 2016) and ROCStories (Mostafazadeh et al., 2016) datasets. Following CommonGen, we conduct part-of-speech tagging over the sentences and further lemmatize the recognized verbs and nouns to obtain the concept sets.

**Baselines** (1) *Concept2Sentence:* We consider several recent submissions to the leaderboard of CommonGen that leverage auxiliary information for GCSR. KFCNet (Li et al., 2021), Re-T5 (Wang et al., 2021), and EKI-BART (Fan et al., 2020) are prototype-based models, which retrieve sentences containing as many input concepts as possible from external captions and NLI datasets, and then use these sentences as auxiliary inputs. VisCTG (Feng et al., 2021b) is an image-augmented model which retrieves images from Google by using concepts as a query, followed by an image captioning model that generates captions as auxiliary inputs. KG-BART (Liu et al., 2020) is a knowledge graph-augmented model which retrieves relations between concepts from ConceptNet as auxiliary inputs. SAPPHIRE (Feng et al., 2021a) is a keyword-based model which extracts keywords from sentences as auxiliary inputs only during training. We also compare to Node2Text, which fine-tunes a pre-trained auto-regressive LM to take the concatenation of concepts as input and output the target sentences. (2) *Concept2Story:* We augment Node2Text with the iterative generation pipeline as in our method, which generates the next sentence given the provided context, previously generated sentences and the current concept set. In addition, we experiment with two representative methods from the controlled text generation literature. Plan-and-write (Yao et al., 2019) first generates storyline keywords, then uses the keywords to generate a story. We use the concept set and context to generate storyline keywords. Action-Plan (Fan et al., 2019) uses predicate-argument pairs as storyline. We adapt the KFCNet model to retrieve prototype sentences. All Concept2Story baselines are used in an iterative generation pipeline, to enable fair comparison to our method.

**Evaluation Metric** We evaluate systems against the $K$ reference sentences provided by a dataset, by measuring the similarities between the machine-generated text and the gold references. Following CommonGen (Lin et al., 2020), we adopt widely-used automatic metrics for evaluating text generation, which focus on (1) n-gram overlap: BLEU (Papineni et al., 2002), ROUGE (Lin, 2004), and METEOR (Banerjee & Lavie, 2005), and (2) concept association: CIDEr (Vedantam et al., 2015) and SPICE (Anderson et al., 2016). Lin et al. (2020) reports that SPICE yields the best correlation with human judgments and thus we used it as the main evaluation metric.

Table 2: Performance comparison with the top-ranked, published models on the official CommonGen test set. [*]Note that KFCNet uses a much larger corpora (over 70M) to retrieve prototypes and on average less than one concept in the concept sets is not covered (Li et al., 2021), while we filter out any SKGs that contain concept sets that overlap with CommonGen dataset.

| Model | BLEU-4 | CIDEr | SPICE |
|---|---|---|---|
| KFCNet (Li et al., 2021)[*] | **43.62** | **18.85** | **33.91** |
| RE-T5 (Wang et al., 2021) | 40.86 | 17.66 | 31.08 |
| VisCTG (Feng et al., 2021b) | 36.94 | 17.20 | 29.97 |
| SAPPHIRE Feng et al. (2021a) | 37.12 | 16.90 | 29.75 |
| KG-BART Liu et al. (2020) | 33.87 | 16.93 | 29.63 |
| EKI-BART Fan et al. (2020) | 35.95 | 17.00 | 29.58 |
| T5-base (our implementation) | 33.81 | 15.79 | 28.34 |
| T5-large (our implementation) | 32.85 | 15.76 | 28.38 |
| T5-large (reported) | 31.96 | 15.13 | 28.86 |
| I&V (T5-base) | 40.16 | 17.44 | 30.57 |
| I&V (T5-large) | 40.57 | 17.71 | 31.29 |

## 4 RESULTS AND ANALYSIS

We design experiments to answer the following questions: (1) Does contextualized scene imagination improve the performance of GCSR models? (2) Does imagination allow GCSR models to learn with less data? (3) How does each source of scene knowledge for pretraining affect the GCSR performance? (4) Do generated SKGs make common sense and correspond to the generated text?

### 4.1 MAIN RESULTS

We compare our proposed approach with state-of-the-art text generation methods on two GCSR tasks to understand whether scene imagination helps GCSR. Table 2 shows the performance of dif-

Table 3: Performance of the compared methods on the Concept2Story tasks. Best results are bold-faced. We mark them with an asterisk if they exceed the second best with statistical significance (p-value < 0.05).

| | Concept2Story-VIST | | | | | | Concept2Story-ROC | | | | | |
| | T5-base | | | BART-large | | | T5-base | | | BART-large | | |
| Model | BLEU-4 | CIDEr | SPICE | BLEU-4 | CIDEr | SPICE | BLEU-4 | CIDEr | SPICE | BLEU-4 | CIDEr | SPICE |
|---|---|---|---|---|---|---|---|---|---|---|---|---|
| Node2Text | 20.64 | 25.41 | 58.55 | 18.52 | 22.91 | 55.48 | 23.31 | 29.32 | 57.66 | 20.60 | 26.09 | 53.80 |
| Keyword | 16.75 | 21.87 | 56.23 | 15.62 | 20.86 | 55.49 | 22.24 | 27.05 | 50.41 | 22.14 | 27.40 | 49.52 |
| Action-Plan | 17.84 | 22.77 | 57.11 | 16.20 | 21.10 | 54.77 | 21.15 | 27.32 | 56.14 | 20.45 | 26.29 | 54.32 |
| Prototype | 20.28 | 25.05 | 58.17 | **22.81** | **26.93** | 58.84 | 23.59 | 29.48 | 57.68 | 26.76 | 31.60 | 58.35 |
| I&V | **21.05**$^*$ | **25.78**$^*$ | **59.21**$^*$ | 22.45 | 26.80 | **59.11**$^*$ | **26.77**$^*$ | **32.33**$^*$ | **60.63**$^*$ | **28.30**$^*$ | **33.40**$^*$ | **60.39**$^*$ |

ferent models on CommonGen. We have the following observations. First, I&V drastically improves the vanilla T5-large model (Node2Text), demonstrating the effectiveness of the imagination module in GCSR. We also provide concrete examples in §A.4 which showcase how imagination fixes errors made by Node2Text. All these errors can be attributed to the fact that Node2Text does not properly capture the commonsense relations between concepts while our imagination module learns how concepts are related from indirect supervision. Second, our model outperforms other models using different auxiliary inputs, including prototypes (Re-T5 and EKI-BART), knowledge facts (KG-BART) and images (VisCTG), showing the benefit of SKGs over these knowledge sources. Although our model under-performs KFCNet, our analysis in their work reveals that 97.4% of the test cases have perfectly matched prototypes, i.e., sentences containing all the queried concepts. It is thus unclear whether KFCNet is conducting commonsense reasoning or merely rephrasing the prototypes. Note that we filter out any collected SKGs that cover the concept sets from the downstream datasets. This ensures that the imagination module is examined with its compositional generalization.

Table 3 shows the experimental results by I&V on the two Concept2Story datasets using T5-base and BART-large as the backend respectively. Among most evaluation metrics, our method outperforms Node2Text and baselines with other intermediate representations incorporated in the same backends. This demonstrates that our imagination module can provide contextualized scene imagination that are more helpful in guiding long narrative generation.

## 4.2 Performance Analysis

**How does the knowledge source affect GCSR?** We perform an ablation study in order to understand how effectively each source of SKGs contributes to the imagination. Specifically, we use each of the following SKG sources to pre-train an imagination module using T5-large as the backend: the silver-standard SKGs extracted from the training set from the downstream task (Task-AMR), and the external SKGs: Caption-AMR, Story-AMR, and VG-SceneGraph (§2.2). For CommonGen, we do not further fine-tune the imagination module in order to distinguish the contributions from each knowledge source more clearly. For Concept2Story (ROCstories), we conduct further fine-tuning using the task-AMR. Since this task provides the context as input, we find it helpful to adapt the imagination module with the task dataset.

The results are shown in Table 4 and we have the following observations. For CommonGen, the contribution comes mostly from the SKGs based on Caption-AMR while being less from VG-SceneGraph. This may due to the fact that VG-SceneGraph is biased towards spatial relations and attributes of objects. For Concept2Story, we find both Story-AMR and Caption-AMR to be helpful for continual pretraining. The former teaches the model to generate contextualized imagination which is necessary for story generation in particular while the latter teaches the model about general commonsense knowledge. For both datasets, the imagination modules that are pre-trained over all the SKG instances yield significantly better results than the ones trained on the task-AMR datasets. This validates our intuition that different sources of SKGs contain complementary commonsense knowledge, and they should be used together for machine imagination.

**How does the backbone LM size affect the module's performance?** We also ablate the LM architecture of the imagination module and the verbalization module respectively to see how our method work with different pre-trained LMs. For the imagination module, we use T5-base and T5-large. This is to investigate how the capacity of LMs affects the learning of scene knowledge.

Table 4: Performance of our method using different SKG sources to train the imagination module, with T5-large as the backbone LM.

| Knowledge Source | CommonGen (in-house) | | | Concept2Story-ROC | | |
|---|---|---|---|---|---|---|
| | BLEU-4 | CIDEr | SPICE | BLEU-4 | CIDEr | SPICE |
| Task-AMR | 28.87 | 15.74 | 31.22 | 23.14 | 29.25 | 57.91 |
| Caption-AMR | 32.21 | 16.14 | 32.16 | 23.77 | 29.76 | 58.46 |
| Story-AMR | 23.73 | 13.51 | 27.53 | 24.17 | 30.10 | 58.59 |
| VG-SceneGraph | 21.00 | 13.36 | 29.07 | 22.84 | 25.33 | 53.96 |
| All-SKG | **33.27** | **16.95** | **33.49** | **26.77** | **32.33** | **60.63** |

Table 5: SPICE performance of our method using different sizes of T5 as backbone for the imagination module.

| Dataset / Backbone LM | **T5-base** | **T5-large** |
|---|---|---|
| CommonGen (in-house) | 32.00 | 33.49 |
| Concept2Story-ROC | 59.56 | 60.63 |

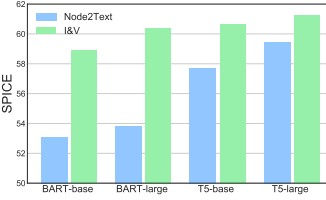

Figure 4: Ablation study on backbone LM sizes of our verbalization module and Node2Text using the Concept2Story-ROC dataset.

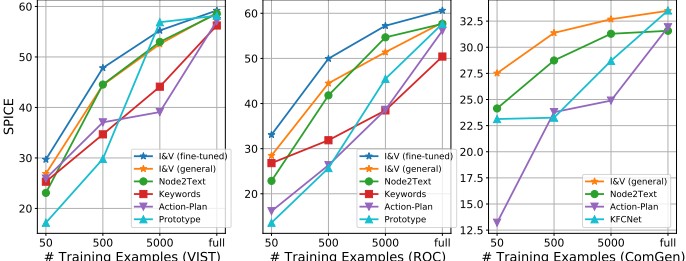

Figure 5: Results (SPICE) of the low-resource experiment on the three benchmark datasets with different number of training examples.

The results are shown in Table 5. Compared to T5-large, we observe a slight performance drop for T5-base, which indicates that larger LMs are able to encode our rich set of SKG instances in a more expressive manner. For the verbalization module, we use BART-base/large and T5-base/large. The results are shown in Figure 4. We observe that compare to baseline, our method consistently yields a better performance regardless of what LM architecture is used.

**Does imagination allow models to learn (faster) with less data?**    Next, we study how the indirect supervision provided to the imagination module help the system effectively learn with limited task-specific training data. Accordingly, we conduct a low-resource experiment where we randomly sample $\{50, 500, 5000\}$ training and development examples from each dataset. For each data size, we use 5 random seeds to obtain 5 different training and development splits. On each split, we train and test with random initialization of 3 seeds, and we report the average on the total 15 ways of results. In this study, the imagination module is fixed untrainable after continual pretraining and is not fine-tuned over the sampled task datasets.

Figure 5 shows that our model consistently outperforms the baselines, and the performance gain is larger when less training data are used. This indicates that rich sources of SKGs provide practical forms of indirect supervision to complement limited task-specific training data. The robustness of our model in low-resource settings also justifies the need for including contextualized SKGs as an intermediate representation, which further enhances the verbalization module to generate plausible sentences even with little training data.

**Is context helpful for imagination?**    To validate that the textual context, including the provided context as well as the previously generated sentences, is helpful for imagination in the Concept2Story task, we conduct an ablation study where we learn an uncontextualized imagination module which only takes concepts as input. The final results on VIST and ROC datasets are 47.32 and 45.18 (SPICE) respectively, which are much lower than the results from contextualized I&V (59.21 and 60.63). This demonstrates that the context is critical in generating SKGs which are more relevant to the story line and thus lead to better text generation.

## 4.3 HUMAN EVALUATION ON GENERATED SKGS

We conduct human evaluation on the SKGs generated by our imagination module to examine their quality. Annotators are presented with the input concepts, the generated SKGs, the predicted sen-

tences resulting from the corresponding SKGs and the ground-truth sentences for reference. For each dataset, 100 instances are randomly chosen for evaluation. Annotators are students majoring in computer science and not all of them know about AMR language prior to the human evaluation. To facilitate annotators' understanding of the evaluation task and AMR, we provide the detailed instruction and the examples of AMR relations. The annotators are asked to judge for: 1) *Completeness*, whether the SKG includes all the concepts (both given and implicit) to constitute a coherent scene; 2) *CommonSense*, whether the SKG organizes the concepts in a way that follows common sense; 3) *Alignment*, whether the generated sentence aligns with the SKG and 4) *Similarity*, whether the predicted sentence is similar to any referenced sentence in semantic. Annotation is based on a 3-point scale: a) *0* – "I do not agree", b) *0.5* – "I partially agree" and c) *1.0* – "I fully agree".

Table 6 shows the evaluation results where we get a fair level of agreement measured by Fleiss Kappa ($\kappa = 0.21$). We observe that the generated SKGs are complete and follow common sense in a high degree across three datasets, which demonstrates the effectiveness of learning useful commonsense knowledge with vast indirect supervision from different resources. Moreover, the SKGs are well-aligned with the generated text, which indicates that the verbalization module consistently follows the guidance of the imagination module when generating sentences. The moderate similarity scores validate that the generated text is generally similar to the natural language sentences annotated by humans.

Table 6: Human evaluation on the generated SKGs regarding *Completeness* (COM), *CommonSense* (CS) and *Alignment* (AL) and *Similarity* (SIM).

|  | COM | CS | AL | SIM |
|---|---|---|---|---|
| CommonGen | 97.30 | 90.15 | 89.90 | 88.30 |
| VIST | 93.80 | 89.70 | 91.40 | 76.20 |
| ROC | 95.70 | 86.60 | 87.80 | 75.68 |

## 5 RELATED WORK

**Knowledge-Enhanced GCSR**  Recent works (Liu et al., 2020; Li et al., 2021) on GCSR propose to retrieve external knowledge to enhance the text generation. Prototype-based models, including EKI-BART (Fan et al., 2020), Re-T5 (Wang et al., 2021), and KFCNet (Li et al., 2021) retrieve massive prototype sentences from external corpora like visual captions and Wikipedia as auxiliary input to the LM. Though the retrieved prototype sentences provide high coverage on the concepts, their model is supervised to compose sentences that are very similar to those existing prototypes. It is thus unclear whether their models are conducting commonsense reasoning or only mimicking the prototypes. KG-BART (Liu et al., 2020) incorporates the embedding of relational facts about the concepts from ConceptNet into both the encoders and decoders of the BART architecture (Lewis et al., 2020). As there could be multiple relations between two concepts, it is unclear how to select the relation that fits a given context (Fadnis et al., 2019). Our imagination module infers the relations between concepts by taking all the concepts into consideration and organizes them in a coherent way.

**Content Planning**  Our method is also related to prior works (Goldfarb-Tarrant et al., 2020) that propose intermediate representations as a way to "plan ahead" before generating long narratives. Plan-and-write (Yao et al., 2019) generates chains of keywords as a storyline, but do not consider relations between keywords (concepts) as we do. Action-plan (Fan et al., 2019) takes a step further by using predicate-argument with semantic role labeling, but still does not involve all the concepts in a sentence. Moreover, these methods are limited to obtaining supervision from task-specific datasets, while we gather effective indirect supervision signals from rich multi-source, multi-modal SKG representations without the need for additional annotations.

## 6 CONCLUSIONS

This paper proposed to enhance neural architectures for GCSR with an intermediate imagination layer. We divided the GCSR process into two steps: imagination, which generated a plausible scene knowledge graph for a given set of concepts, and verbalization, which transformed this scene graph into a fluent sentence that corresponds to human common sense. The method was trained with diverse scene knowledge graphs derived from both text and vision modalities. Our experiments demonstrated the ability of the proposed method to perform GCSR effectively, by describing plausible scenes, and efficiently, by requiring less training data. The image caption graphs proved most beneficial to learn from. Future work should investigate the impact of imagination on interactive commonsense tasks, like dialogue generation, and include scene graphs from the audio modality.

## ACKNOWLEDGMENTS

We thank the anonymous reviewers and all the collaborators in USC INK research lab for their valuable feedback. This material is based upon work sponsored by the DARPA MCS program under Contract No. N660011924033 with the United States Office Of Naval Research.

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

# A APPENDIX

## A.1 RULES FOR MAPPING VISUAL SCENE GRAPHS TO SKG

There are 3858 relation types in our processed VisualGenome dataset due to the noisy annotation. We map these relations into 8 relations. For relations that are annotated as verbs by VisualGenome, we break the relationship (subject, relation, object) into (relation, :ARG0, subject) and (relation, :ARG1, oject). For other popular relations, we conduct the following mapping:(subject, be, object)→(subject, domain, object), (subject, displace, object)→(subject, possible, object),(subject, have/of, object)→(subject, part, object),(subject, with, object)→(subject, poss, object),(subject, on/behind/at/under/along/in/..., object)→(subject, location, object). The remaining relations that do not follow the above mapping criteria are mapped to an "other" relation. Note that the 7 non-"other" relations make up 97.73% of the triplets in VisualGenome.

Table 7: The most common relation types in SKG instances and their example triplets.

| Relation types | Examples |
|---|---|
| ARG1 | (play, ARG1, guitar) |
| ARG0 | (play, ARG0, man) |
| ARG2 | (ask, ARG2, girl) |
| Location | (play, Location, stage) |
| Time | (play, Time, sing) |
| Op1 | (down, Op1, stair) |
| Part | (dog, Part, ear) |

## A.2 IMPLEMENTATION DETAILS

For the main experiments, we develop the imagination module by continually pre-train a T5-large model over the caption, story, and vision SKGs. We then further adapt the imagination module over each task dataset annotated with the silver-standard SKGs by further fine-tuning. To train the verbalization module, we fine-tune T5-base and BART-large as two backend LMs. During training, we use both silver-standard SKGs and generated SKGs, while averaging the training loss associated with each of them. We use the Adam optimizer with weight decay $1e - 2$. We search the optimal hyper-parameters based on the perplexity over the development set, where the learning rate is chosen from $\{2e - 6, 1e - 5, 3e - 5, 1e - 4\}$, the batch size is chosen from $\{8, 16, 32, 64, 128\}$.

## A.3 STATISTICAL ANALYSIS

Table 8: Statistical analysis (p-values) for the ablation study on the backend LM used by the verbalization module and the low-resource experiment. $< 0.01$ indicates a significant improvement and $< 0.05$ indicates a fairly significant improvement, and NA indicates that our method does not outperform the best baseline.

| | BART-base | BART-large | T5-base | T5-base |
|---|---|---|---|---|
| | $< 0.01$ | $< 0.01$ | $< 0.01$ | $< 0.01$ |

| # Training examples | CommonGen (in-house) | VIST | ROC |
|---|---|---|---|
| 50 | | $< 0.01$ | $< 0.01$ | $< 0.01$ |
| 500 | | $< 0.01$ | $< 0.01$ | $< 0.01$ |
| 5000 | | $< 0.01$ | NA | $< 0.01$ |
| All | | NA | $< 0.05$ | $< 0.01$ |

We conduct statistical significance analysis on the experiments involving baselines, which include the ablation study on the backend LM used by the verbalization module (Figure 4) and the low-resource experiment (Figure 5). The p-values are shown in Table 8, where $< 0.01$ indicates a significant improvement and $< 0.05$ indicates a fairly significant improvement, and NA indicates that our method does not outperform the best baseline.

## A.4 QUALITATIVE ANALYSIS ON HOW IMAGINATION HELPS

We show how imagination can help generating sentences that follow common sense via qualitative analysis in Table 9-10. As comparison, we also show the results from the Node2Text baseline which does not imagines. We organize the results based on 5 (not necessarily exclusive) types of errors made by Node2Text, which include incorrect role attribution to 1) agents, 2) actions or 3) objects, 4) failing to infer the implicit concepts and 5) misunderstanding the relations between events.

## A.5 QUALITY EVALUATION ON THE GENERATED SKGS

Since there are no grountruth SKGs annotated in downstream datasets, we use the silver-standard SKGs as reference to give a rough estimation of the quality of the generated SKGs. We focus on re-

Table 9: Qualitative analysis on errors made without imagination and how imagination can help fix the errors (Part 1). The left arrow ⟵ indicates the key relations that fix the errors.

| Error 1 (Incorrect Agent) | Example 1 | Example 2 |
|---|---|---|
| Input concepts | {owner, chase, dog, ball, throw} | {ski, rope, hold, boat, pull} |
| Text w/o imagination | The **dog** is chasing the ball and **throwing** it at the owner. | A woman skis downhill as **she pulls** a boat holding a rope. |
| Text w/ imagination | A dog chases a ball being thrown by its owner. | A boat pulls a skier who is holding a rope. |
| Generated SKG | (chase, ARG0, dog), (chase, ARG1, ball), (throw, ARG1, ball), (throw, ARG0, owner) ⟵ | (pull, ARG0, boat), ⟵ (pull, ARG1, person), (ski, ARG0, person), (hold, ARG0, person), (hold, ARG1, rope) |
| Error 2 (Incorrect Action) | Example 1 | Example 2 |
| Input concepts | {butter, pot, crack, egg, add} | {stand, tongue, stick} |
| Text w/o imagination | She adds eggs, **crackers**, and butter to a pot. | A boy stands next to a **stick** of his tongue. |
| Text w/ imagination | You crack an egg and add butter to a pot. | A man stands with his tongue sticking out. |
| Generated SKG | (crack, ARG0, you), (crack, ARG1, egg), ⟵ (add, ARG0, you), (add, ARG1, butter), (add, ARG2, pot) | (stand, ARG1, man), (man, part, tongue), (stick, ARG0, man), (stick, ARG1, tongue), ⟵ (stick, ARG2, out) |
| Error 3 (Incorrect Object) | Example 1 | Example 2 |
| Input concepts | {hit, bottle, shoe, open, wall} | {wear, talk, phone} |
| Text w/o imagination | Someone opens his **shoe** and hits a **bottle** on the wall. | A woman is wearing a **cell phone** and talking to the camera. |
| Text w/ imagination | A man opens a bottle and hits his shoe against a wall. | A man wearing glasses is talking on the phone. |
| Generated SKG | (open, ARG0, man), (open, ARG1, bottle), ⟵ (hit, ARG0, man), (hit, ARG1, shoe), ⟵ (shoe, poss, man), (hit, ARG2, against), (against, op1, wall) | (talk, ARG0, man), (wear, ARG0, man), (wear, ARG1, glasses), ⟵ (talk, medium, phone) |

call since the silver-standard SKGs may not cover all the plausible scenes. Our evaluation considers the following three metrics. 1) Average recall of the given concepts (Explicit Concepts) to examine whether an SKG contains all the given concepts. 2) Average recall of the implicit concepts (Implicit Concepts) to examine whether an SKG also contains implicit concepts. The implicit concepts for reference are the nodes from the silver-standard SKGs excluding the given concepts. 3) Average recall of the relations (Relation) to examine the proportion of the referenced relations that are covered by the generated SKGs. Here, a relation is considered as correct only if the head concept, relation and the tail concept all match the reference.

The results shown in Table 11 indicate a fairly good quality of the generated SKGs, which connect all the given concepts for over 99% of the cases and have a large overlap (over 68%) with the silver-standard SKGs. Note that the particular low recall on implicit concepts is due to the fact that there can be many different implicit concepts to constitute a complete SKG.

Table 10: Qualitative analysis on errors made without imagination and how imagination can help fix the errors (Part 2). The left arrow ⟵ indicates the key relations that fix the errors.

| Error 4 (Implicit Concepts) | Example 1 | Example 2 |
|---|---|---|
| Input concepts | {fill, liquid, machine, bottle} | {lasso, catch, horse, animal, ride} |
| Text w/o imagination | A machine holding a bottle filled with liquid. | Animals ride a horse that caught a lasso. |
| Text w/ imagination | A **man** holds a bottle filled with liquid in a machine. | A **man** riding a horse to catch an animal with a lasso. |
| Generated SKG | (hold, ARG0, man), ⟵ (hold, ARG1, bottle), (fill, ARG1, bottle, (fill, ARG2, liquid), (hold, location, machine) | (ride, ARG0, man), ⟵ (ride, ARG1, horse), (ride, purpose, catch), (catch, ARG0, man), (catch, ARG1, animal), (catch, instrument, lasso) |
| Error 5 (Event Relations) | Example 1 | Example 2 |
| Input concepts | {trick, perform, begin, stunt, dance} | {stir, pour, pot, ingredient, begin} |
| Text w/o imagination | A group of people begin performing a stunt **while** performing a trick. | She begins stirring the ingredients in the pot and begins pouring them into the water. |
| Text w/ imagination | A man performs stunts and tricks as he begins to dance. | He pours the ingredients into the pot and **begins** to stir them. |
| Generated SKG | (perform, ARG0, man), (perform, ARG1, stunt), (perform, ARG1, trick), (perform, time, begin), ⟵ (begin, ARG0, man), (begin, ARG1, dance), (dance, ARG0, man) | (pour, ARG0, he, (pour, ARG1, ingredient), (pour, ARG2, pot), (begin, ARG0, he), ⟵ (begin, ARG1, stir), (stir, ARG0, he), (stir, ARG1, ingredient) |

Table 11: Quality evaluation (recall) on the generated SKGs with silver-standard SKGs as reference.

| Dataset | Explicit Concepts | Implicit Concepts | Relation |
|---|---|---|---|
| CommonGen (in-house) | 99.81 | 17.07 | 72.10 |
| VIST | 99.96 | 35.66 | 68.19 |
| ROC | 99.95 | 61.86 | 74.04 |

## A.6 ABLATION STUDY ON INPUT TO I&V

For imagination, the inclusion of context helps the module to generate contextualized SKG which is more relevant to the current storyline. To justify this design choice, we conduct an ablation study where we learn an uncontextualized imagination module which only takes concepts as input. The resulting SPICE scores are 47.32 and 45.18 on VIST and ROC datasets respectively, which are much lower than the results from contextualized I&V (59.21 and 60.63 in SPICE respectively). This demonstrates that the context is critical in generating relevant SKGs which lead to better text generation.

For verbalization, the textual context is important for narrative generation in keeping the storyline consistent. The concept input helps indicate what are the nodes while the SKG input is about the edges. We conduct an ablation study on what input to include for verbalization. The results in Table 12 show that adding concepts as input generally helps improve the performance of our method while adding context is critical for story generation.

Table 12: Ablation study on what input is fed to the verbalization module.

| Input | CommonGen (in-house) | VIST | ROC |
|---|---|---|---|
| SKG-only | 33.39 | 17.13 | 18.90 |
| Concept + SKG | **33.49** | 27.01 | 36.42 |
| Context + SKG | NA | 57.99 | 58.06 |
| Context + Concept + SKG | NA | **59.21** | **60.63** |

Table 13: Ablation study on using 1) silver-standard SKGs, 2) generated SKGs or 3) both to train the verbalization module.

| Input SKGs | CommonGen (in-house) | VIST | ROC |
|---|---|---|---|
| Silver-standard | 33.19 | 53.26 | 60.55 |
| Generated | 32.56 | 58.34 | 59.82 |
| Silver. + Generated | **33.49** | **59.21** | **60.63** |

Table 14: Ablation study on the design of the generation process.

| Generation Process | VIST | ROC |
|---|---|---|
| All-at-once | 57.16 | 54.83 |
| Independent | 27.01 | 36.42 |
| Iterative (I&V) | **59.21** | **60.63** |

## A.7 ABLATION STUDY ON WHAT SKGS TO USE WHEN LEARNING VERBALIZATION

We conduct an ablation study where we use 1) silver-standard SKGs only, 2) generated SKGs only and 3) both types of SKGs during training the verbalization module. The results in Table 13 validates that using both types of SKGs yield to the best performance of our method.

## A.8 ABLATION STUDY ON CONCEPT-DROPOUT

We conduct an ablation study where we do not drop any concepts when we train the imagination module. We then apply the imagination module on CommonGen and conduct the experiments. The final performance is 28.28 in SPICE while the system with the imagination module trained with concept-dropout achieves 33.49. This validates that dropping concepts is necessary since in the downstream tasks not all the concepts are provided and the model needs to infer the implicit ones.

## A.9 ABLATION STUDY ON THE GENERATION PROCESS

We conduct an ablation study to verify that the iterative generation process is more effective than 1) generating all the sentences at once, and 2) generating each sentence independently. For baseline 1), we learn an uncontextualized imagination module which only takes concepts as input and does not need the previous generated context. We apply the uncontextualized imagination module to generate all the SKGs at once and then the verbalization module generates all the target sentences at once by taking the provided context, all the concept sets and all the SKGs as input. For baseline 2), we still use the contextualized imagination module to generate an SKG at a time. But we learn a verbalization module which does not take the previously generated sentences as input and thus generate each target sentence independently. We conduct the ablation study on the two datasets of the concept2story task (we do not consider the CommonGen benchmark here since there is only one target sentence to be generated in CommonGen). The results of the average SPICE scores from 3 runs are shown in Table 14. The two baselines are both outperformed by our iterative approach, which verifies that the previously generated sentences are important for both imagination and verbalization and thus the iterative generation process is necessary.

Table 15: Evaluation results (SPICE) with I&V using silver-standard SKGs during inference.

| SKGs (inference) | CommonGen (in-house) | VIST | ROC |
|---|---|---|---|
| Silver-standard | 41.85 | 69.34 | 67.70 |
| Generated | 33.49 | 59.21 | 60.63 |

## A.10 EXPERIMENTS WITH SILVER-STANDARD SKGS DURING INFERENCE

We report the "oracle" performance of our system using silver-standard SKGs during inference to estimate the upper-bound of our method. The result is shown in Table 15.

