# OpenReview forum: "Contextualized Scene Imagination for Generative Commonsense Reasoning"
_ICLR.cc/2022/Conference — ICLR 2022 Poster_

### Official Review · Reviewer_GjaV · 2021-11-01

**Correctness:** 4
**Technical Novelty And Significance:** 3
**Empirical Novelty And Significance:** 4
**Recommendation:** 8
**Confidence:** 4

**Main Review:**

**Strengths**
The task of generative commonsense reasoning is interesting but challenging. I like the idea of decomposing imagination and verbalization. The idea of using graphs as better representations (compared to just sets), and the idea of leveraging graph annotations from various modalities make a lot of sense to me. The paper is well written, the experiments are well designed.

The results are convincing, I appreciate the authors providing significance test in multiple tables, and details about their human evaluation. "We conducted this study with 3 individuals who are studying Computer Science, though not all individuals were aware of SKG’s prior to the human evaluation." this is important information to me.

**Questions and concerns**
1. In Section 2.3, the authors mention that they randomize the order of concepts during training, does this really help in the GCSR tasks, for instance, compared to using the concept order provided by the dataset (or maybe other ordering strategies, like to sort the concepts so that the model always observe concepts in a consistent order). The concern comes from the question of, given a data point, are the concepts really a *set* or a *list*. It may depend on how the data is collected, for example, humans may have some common preferences extracting keywords from a data source (e.g., a story, an image).
2. In the same paragraph, the authors mention that they randomly "dropout" a subset of the concepts to encourage generalization. Do the authors provide experiments discussing this?
3. On the top of Section 3, the authors say that during the training of the verbalization module, they use both 1) silver-standard SKG and 2) imagination module-generated SKGs as input. Are there ablation studies on each of these two?
4. In addition to the AMR tree, did the author try other semantic parsing methods, e.g., semantic role labeling? Do the authors expect to see similar improvement if use AMR?
5. What are the evaluation scores if the system is equipped with silver-standard SKGs? I.e., testing the verbalization module with SKGs returned by a parser? Comparing this number with Table 2 and Table 3 may make the motivation of having a neural imagination module stronger.
6. It would be great if a few examples of the model outputs/behaviors, or other qualitative analyses can be provided.
7. (minor) AMR parsing will specify words (e.g., verbs) with their word sense. For instance, the verb "want" will be something like "want-01" in the tree. How do the authors deal with these?
8. (minor) In appendix A.2, the authors describe details about human evaluation, I appreciate it. To me, I would also see a model's surprising score (sense-making output which being absent from the given concept set) since one of the motivations of using an imagination module is to "imagine".
9. (minor, and maybe a bad question) Why not trying graph neural networks to encode/represent SKGs? What extra ability does the Language Model in verbalization module enable.


**Summary Of The Paper:**

In this work, the authors propose a novel framework for the generative commonsense reasoning (GCSR) tasks. GCSR are tasks where given a set of concepts (or a sequence of set of concepts), a model is required to generate sentences (or a short paragraph) that are both grammatically correct and plausible (follow commonsense).

The authors propose Imagine-and-Verbalize (I&V), which is a 2-layer system:
1. The imagination module is a transformer-based seq2seq model, takes a set of concepts (and sometimes text as context) as input, and generates a flattened graph (AMR tree, PENMAN serialization).
2. The verbalization module is another transformer-based seq2seq model, which takes the concepts, some context, and the generated scene knowledge graph (SKG, flattened) as input, and translate the scene information into natural language sentences (or iteratively, paragraphs).

On two GCSR datasets, the proposed method significantly outperform strong baselines and prior works.

**Summary Of The Review:**

Overall I believe this is a good paper, to me it is certainly above the bar. While I think the paper can be further improved (see my questions above), I would suggest to accept this paper.

But of course, correct me if I misunderstood the paper.

---

> ### Author Response · Authors · 2021-11-18
> **Response to Reviewer GjaV (Part 1)**
>
> Thank you for your positive review and very helpful comments! Below please find our responses to your questions. We have also updated the draft accordingly.
>
> >1.Does randomizing concepts during training really help in the GCSR tasks? Is the concepts set really a set instead of a list?
>
> Thanks for this insightful question! In both CommonGen and concept-to-story datasets, the order of concepts in the train/test sets is **randomized** in order not to leak the information about how they are ordered in the ground-truth sentence. So training our imagination and verbalization modules to be invariant to a specific order is necessary. We have an ablation study where we sort the concepts by their alphabetical order during both training and inference. This does not lead to better performance compared with our strategy. The results are shown below.
>
> | Order      | CommonGen | VIST | ROC |
> | -----------  | -----------------: | ------: | ------: |
> | Sorted    |              31.80 | 38.28 | 43.94 |
> | Random | **33.49** | **59.21** | **60.63** |
>
> >2. The authors mention that they randomly "dropout" a subset of the concepts to encourage generalization. Do the authors provide experiments discussing this?
>
> Yes. We conducted an ablation study where we do not drop any concepts when we train the imagination module. We then apply the imagination module on CommonGen and conduct the experiments. The final performance is 28.28 in SPICE while the system with the imagination module trained with concept-dropout achieves 33.49. This validates that dropping concepts is necessary since in the downstream tasks, not all the concepts are provided and the model needs to infer the implicit ones. We have included this discussion in Appendix A.8.
>
> >3. During the training of the verbalization module, they use both 1) silver-standard SKG and 2) imagination module-generated SKGs as input. Are there ablation studies on each of these two?
>
> Yes. We have an ablation study where we use 1) silver-standard SKGs only, 2) generated SKGs only and 3) both types of SKGs during training. The results shown below validates that using both types of SKGs yields the best performance of our method. We have also included this ablation study in Appendix A.7.
>
> | Training SKGs  | CommonGen | VIST | ROC |
> | -------------------- | -----------------: | ------: | ------: |
> | Silver-standard |             33.19 | 53.26 | 60.55 |
> | Generated        |             32.56 | 58.34 | 59.82 |
> | Both                  | **33.49** | **59.21** | **60.63** |
>
> >4. Did the author try other semantic parsing methods, e.g., semantic role labeling? Do the authors expect to see similar improvement if they use AMR?
>
> The action-plan baseline we consider in the concept-to-story task uses predicate-arguments based on SRL as the intermediate representation. The experiments on both VIST and ROC datasets (Table 2) show that SKG is a more effective representation of scene knowledge.
>
> >5.What are the evaluation scores if the system is equipped with silver-standard SKGs?
>
> We report the “oracle” performance of our system using silver-standard SKGs during inference by viewing silver-standard SKGs as oracle graphs. The results are also reported in Appendix A.10.
>
> | SKGs (inference) | CommonGen | VIST  | ROC  |
> | ----------------------- | -----------------: | -------: | ------: |
> | Silver-standard    |              41.85 | 69.34 | 67.70 |
> | Generated           |              33.49 | 59.21 | 60.63 |

---

> > ### Author Response · Authors · 2021-11-18
> > **Response to Reviewer GjaV (Part 2)**
> >
> > >6. Provide a few examples of the model outputs/behaviors.
> >
> > We summarize how imagination can help improve the commonsense aspect of the generated text with examples in Appendix A.4. As a comparison, we also show the results from the Node2Text baseline which does not leverage imagination. We organize the results based on 5 (not necessarily exclusive) types of errors made by Node2Text, which include incorrect role attribution to 1) agent (*dog* instead of *person* throwing a ball), 2) action (a *stick* of tongue instead of *sticking out* his tongue) or 3) object (wearing *phone* instead of *glasses*), 4) failing to infer the implicit concepts and 5) misunderstanding the relations between events. Below shows two examples of the errors and you can find more examples in Appendix A.4.
> >
> > - Example 1 (incorrect agent)
> >
> > Input concepts: {ski, rope, hold, **boat**, pull}
> >
> > Prediction w/o imagination: A woman skis downhill as she pulls a boat holding a rope.
> >
> > Prediction w/ imagination: A boat pulls a skier who is holding a rope.
> >
> > Generated SKG: (pull<:ARG0> boat), (pull<:ARG1> person), (ski<:ARG0> person), (hold<:ARG0> person), (hold<:ARG1> rope)
> >
> > - Example 2 (incorrect object)
> >
> > Input concepts: {wear, talk, **phone**}
> >
> > Prediction w/o imagination: A woman is wearing a cell phone and talking to the camera.
> >
> > Prediction w/ imagination: A man wearing glasses is talking on the phone.
> >
> > Generated SKG: (talk<:ARG0> man), (wear<:ARG0> man), (wear<:ARG1> glasses), (talk<:medium> phone)
> >
> > >7. AMR parsing will specify words (e.g., verbs) with their word sense. How do the authors deal with these?
> >
> > We discard such information since the relations between concepts already reveal the word sense. For example, a concept being the head entity of ARG-X is usually a verb, such as (throw, ARG0, person) or (throw, ARG1, ball).
> >
> > >8. I would also see a model's surprising score (sense-making output which being absent from the given concept set) since one of the motivations of using an imagination module is to "imagine"
> >
> > We agree that it would be interesting to ask human annotators to categorize whether the imagined scene is surprising, trivial, or somewhere in between. We would leave such an investigation as the future extension of the work.
> >
> > >9. Why not try graph neural networks to encode/represent SKGs? What extra ability does the Language Model in the verbalization module enable?
> >
> > We did try using a graph transformer for encoding SKG in our preliminary study and found it difficult to obtain meaningful results. The challenge comes from the structural gap between a graph encoder and a text decoder. The benefit of using a pre-trained LM is that the representation of the encoder and decoder remains in the same semantic space already. In addition, the tokenization technique used by LM provides better generalization to unseen concepts.

---

> ### Comment · Reviewer_GjaV · 2021-11-18
> **Thank You**
>
> I appreciate the authors to answer our questions. I have read other reviewers' comments and the authors' rebuttal. In my opinion the authors did a good job providing supporting information (especially to answer Reviewer 6Z6c's questions). I will keep my score of 8.

---

### Official Review · Reviewer_y2yo · 2021-11-02

**Correctness:** 3
**Technical Novelty And Significance:** 3
**Empirical Novelty And Significance:** 3
**Recommendation:** 8
**Confidence:** 3

**Main Review:**

Overall this is an interesting paper, which shows convincing improvement over most baselines, and which seems to show more efficient learning relative to all selected baselines. The writing of the paper is on the whole clear, and the reasoning behind the use of the intermediate representations is logical and intuitive.

There are, however, a few things that remain a bit unclear, making it more difficult to assess exactly how much and why the imagination/SKG component contributes. First, I'm not seeing a description of what is meant by the "vanilla T5-Large model", though the improvement over that baseline is used as key evidence in favor of the importance of the imagination module. Does this baseline involve T5-Large being fine-tuned to map directly from concepts to sentences? Does it include the textual context input as well? Understanding these details more clearly will help in assessing what exactly has been demonstrated by the improvement over that baseline.

Along a similar line, I'm curious about the inclusion of the concepts and context as input both for the imagination module and for the verbalization module. Why is it necessary to include these at both stages? What does the performance look like if only the SKG is given as input to the verbalization module?

More generally, while it is nice to see the ablations on the specific data types and base model sizes, it seems that it would be more relevant to see some analyses that shed light on how exactly the intermediate representations contribute to improving the model performance. Perhaps an analysis of errors made in the absence of the imagination module that are not made in the presence of the imagination module, or something of this kind. I'm also wondering about the fact that the improvements over some of the strongest baselines are fairly small -- do we think that those models are learning something like these SKGs, but simply doing so in a different (potentially less efficient) way? Discussion of these considerations would improve the clarity of what is being contributed by the model's innovations and why.

Typos / smaller notes:

p3 Pre-trainined
p8 moperformance
Fig 5 "I&V" vs "Imagination" label?

**Summary Of The Paper:**

This paper proposes a model for generative commonsense reasoning, taking a set (or sets) of concepts as input and producing a sentence (or sentences) as output. The primary innovation introduced in this model is the use of an intermediate "scene knowledge graph" (roughly equivalent to an AMR of the corresponding sentence) which is output by an "imagination module" that takes concepts (and textual context) as input. This SKG is then input, along with the concepts and textual context, to a "verbalization module" which outputs a sentence (or sentences). The authors show that this model outperforms most of the selected baselines (on tasks of generating single sentences as well as stories), with the exception of KFCNet, which the authors argue is trained on prototypes from a much larger corpus, and is supervised to produce sentences like these prototypes, possibly limiting the model's need to generalize at test. They also do ablations on the type of training data, and the size of base LM, as well as showing that their model learns better with smaller amounts of training data.

**Summary Of The Review:**

Overall the paper is interesting and mostly clear, and it shows improvement over most baselines, with a fairly convincing argument that it is a more efficient learner with smaller amounts of training data. However, I have some lingering questions about what exactly has been demonstrated in terms of importance of the imagination module, and why certain inputs were included in the verbalization module -- and I would have liked to see more analysis discussion on exactly why the SKGs are improving performance, and how we can think about the way this compares to strong alternative models.

---

> ### Author Response · Authors · 2021-11-18
> **Response to Reviewer y2yo (Part 1)**
>
> Thank you for your positive review and very helpful comments! We will carefully fix the typos and improve the clarity of writing regarding the unclear parts that are pointed out. Below are our responses and we have updated the draft accordingly.
>
> >1. I'm not seeing a description of what is meant by the "vanilla T5-Large model". Does the vanilla baseline involve T5-Large being fine-tuned to map directly from concepts to sentences? Does it include the textual context input as well?
>
> Sorry for the confusion. The vanilla T5-large baseline (Node2Text) takes the context, the previously generated sentences, and the given concepts as input, which is formatted in the same way as in our verbalization module (Figure 2). The previously generated sentences are from the ground-truth during training but are predicted by the model during inference. We fine-tune a T5-large model to learn to map from the above input to the ground-truth output sentence. Details of this baseline are included in Sec 3 of the paper. All the baselines considered in the concept-to-story task adopt the iterative generation paradigm. This makes sure the improvement brought by our method is from the imagination as a superior intermediate representation.
>
> >2. I'm curious about the inclusion of the concepts and context as input both for the imagination module and for the verbalization module. Why is it necessary to include concepts and context at both stages? What does the performance look like if only the SKG is given as input to the verbalization module?
>
> We conducted ablation studies for both the imagination and the visualization modules, and found that context benefits both the imagination module and the verbalization module.
>
> - For imagination, the inclusion of context helps the module to generate contextualized SKG for the concept2story task, which is more relevant to the storyline generated thus far (note that there is no context for the concept2sentence task). As an ablation study, we learn an uncontextualized imagination module which only takes concepts as input. The results of the average SPICE score from three trials are shown below, which demonstrates that the context is critical in generating relevant SKGs which lead to better text generation.
>
> | Imagination Module | VIST | ROC |
> | -------------------------- | ------: | ------: |
> | Uncontextualized    | 47.32 | 45.18 |
> | Contextualized        | **59.21** | **60.63** |
>
> - For verbalization, the textual context is important for narrative generation in keeping the storyline consistent. The concept input helps indicate what are the nodes while the SKG input is about the edges. We conduct an ablation study on what input to include for verbalization. The results show that adding concepts as input generally helps improve the performance of our method while adding context is critical for story generation.
>
> | Input                                   | CommonGen | VIST | ROC    |
> | ----------------------------------- | ----------------: | -------: | --------: |
> | SKG-only                            |              33.39 | 17.13 | 18.90 |
> | Concept + SKG                  |       **33.49** | 27.01 |  36.42 |
> | Context + SKG                   |                  NA | 57.99 | 58.06 |
> | Context + Concept + SKG |       NA | **59.21** | **60.63** |

---

> > ### Author Response · Authors · 2021-11-18
> > **Response to Reviewer y2yo (Part 2)**
> >
> > >3. It would be more relevant to see some analyses that shed light on how exactly the intermediate representations contribute to improving the model performance. Perhaps an analysis of errors made in the absence of the imagination module that are not made in the presence of the imagination module.
> >
> > Thank you for this good suggestion! We characterize the errors made by the Node2Text baseline which does not leverage imagination into 5 categories (not necessarily exclusive), which include incorrect role attribution to 1) agent (*dog* instead of *person* throwing a ball), 2) action (a *stick* of tongue instead of *sticking out* his tongue) or 3) object (wearing *phone* instead of *glasses*), 4) failing to infer the implicit concepts, and 5) misunderstanding the relations between events. All these mistakes result from the absence of an imagination module so that the baseline composes an unusual scene that violates common sense. We show two examples of the errors below and please find more examples in Appendix A.4.
> >
> > - Example 1 (incorrect agent)
> >
> > Input concepts: {ski, rope, hold, **boat**, pull}
> >
> > Prediction w/o imagination: A woman skis downhill as she pulls a boat holding a rope.
> >
> > Prediction w/ imagination: A boat pulls a skier who is holding a rope.
> >
> > Generated SKG: (pull<:ARG0> boat), (pull<:ARG1> person), (ski<:ARG0> person), (hold<:ARG0> person), (hold<:ARG1> rope)
> >
> > - Example 2 (incorrect object)
> >
> > Input concepts: {wear, talk, **phone**}
> >
> > Prediction w/o imagination: A woman is wearing a cell phone and talking to the camera.
> >
> > Prediction w/ imagination: A man wearing glasses is talking on the phone.
> >
> > Generated SKG: (talk<:ARG0> man), (wear<:ARG0> man), (wear<:ARG1> glasses), (talk<:medium> phone)
> >
> > >4. Small improvement over the baselines. Are these models also learning something like SKGs but in a more efficient way?
> >
> > The baselines might be able to learn implicit scene representation as our system does but we do not think the learning process (based on the task training data) is efficient. This is because there are too many (subtle) variances in natural language and it will be difficult to learn how to generate a reasonable sentence by relying on the (concepts, text) pairs alone as we mentioned in Sec 2.1. Our imagination adopts SKG as a high-level representation of a plausible scene which abstracts away from surface forms and it provides the model with denser supervision on how to organize concepts coherently compared with sparse text. Moreover, the SKG representation allows us to collect indirect supervision from a diverse set of resources, which provides a more comprehensive distribution of plausible scenes as detailed in Sec 2.2. As evidenced by the low-resource experiment in the last study of Sec 4.2, our method is even more effective in teaching the generative commonsense reasoning ability to machines when task training data is limited.

---

> > > ### Author Response · Authors · 2021-11-20
> > > **Looking forward to hearing from you**
> > >
> > > Dear Reviewer y2yo,
> > >
> > > We want to send you a friendly reminder for the discussion. Here is a summary of our response for your valuable feedback!
> > >
> > > - We recapped the major contributions in the general response to help you better distinguish our work from prior works.
> > > - We clarified that the vanilla baseline also takes the textual context as input to help you better understand how our method improves over the baseline with imagination.
> > > - We categorized the errors made in the absence of imagination and how those errors are fixed in the presence of the imagination with concrete examples.
> > >
> > > We thank you again for your valuable comments, and we would appreciate it if you could reconsider the evaluation of our work based on our response. We are happy to extend our response if you have any other concerns left.
> > >
> > > Thanks.

---

> > > > ### Comment · Reviewer_y2yo · 2021-11-21
> > > > **Thank you and follow-up**
> > > >
> > > > Thank you for the detailed responses. I appreciate the clarifications, and the addition of new results to the appendices. For the phrase "vanilla T5-Large model": if that is being referred to as "Node2Text" elsewhere, this should be clarified/made consistent. This may be as simple as adding a parenthetical referencing Node2Text after mentioning the "vanilla T5-Large model", but it looks like Table 2 also does not reference "Node2Text", so that is probably adding to the confusion between the description of the baselines and the description of the results.
> > > >
> > > > I appreciate the results that have been added to the appendix, and I think that they can be beneficial to the overall contribution. However, at the moment my understanding is that these results are not referenced at all in the main text, and are not making any contribution to the primary narrative. Preferably, I would like to see at least brief pointers to these components of the appendix in relevant sections of the main paper, so that readers know that these bases have been covered. Additionally, for instance in the case of the error analysis, it would be nice for the takeaways of this analysis to make it into the main paper, so that they can strengthen the central narrative.
> > > >
> > > > Trusting that the authors will make versions of these additional revisions as appropriate, I've raised the score in my review.

---

> > > > > ### Author Response · Authors · 2021-11-22
> > > > > **Appreciate your re-evaluation and our revisions**
> > > > >
> > > > > Thank you so much for your re-evaluation of our work! We have made the following revisions to the paper based on your suggestion.
> > > > > - We added a pointer to the ablation study on what inputs to include for the verbalization module in Sec 2.4.
> > > > > - We clarified that the vanilla T5-large model is the Node2Text in Sec 4.1
> > > > > - We added discussion on the qualitative analysis in Sec 4.1.
> > > > > - We added the ablation study on whether to include context for the imagination module in Sec 4.2.

---

### Official Review · Reviewer_9SQy · 2021-11-02

**Correctness:** 4
**Technical Novelty And Significance:** 3
**Empirical Novelty And Significance:** 3
**Recommendation:** 6
**Confidence:** 4

**Main Review:**

Strengths
- The authors propose a novel imagine-and-verbalize framework: an imagination module learns to construct a contextualized SKG from input concepts; a verbalization module learns to faithfully realize the imagined SKG into natural language. The scene imagination module constructs a structured representation of a plausible scene and formalizes the background knowledge required for the reasoning.
- The authors collect and harmonize knowledge resources from different domains and modalities, providing a rich auxiliary supervision signal for scene imagination.
- The experiments on both concept-to-sentence and concept-to-story generation tasks demonstrate the effectiveness of the proposed framework.

Weaknesses
- The novelty of the whole method may be limited. On the one hand, previous work has used SRL as an intermediate representation, and this work simply replaces it with AMR. On the other hand, pretrained language models are directly adopted in the two stages of the proposed framework. Therefore, there is no innovation in technology.
- The proposed method needs a good AMR parsing tool to get AMR graphs. But training the AMR tool requires expensive annotation data. Does the author check the parsing quality of the parsing AMR tool?
- The evaluation of the generated AMR of the first stage was not given in the experiment. Will the generated AMR graph contain a lot of noise?
- In human evaluation, the authors should also present results of baselines to better demonstrate the improvement of the proposed approach over the baselines.


**Summary Of The Paper:**

Descriptive sentences about arbitrary concepts generated by neural text generation models are often grammatically fluent but may be not consistent with common sense. This show that such generative commonsense reasoning (GCSR) skills are lacking in state-of-the-art text generation methods. Therefore, the authors propose an Imagine-and-Verbalize (I&V) method, which learns to imagine a relational scene knowledge graph (SKG) with relations between the input concepts, and leverage the SKG as a constraint when generating a plausible scene description. Experimental results demonstrate the effectiveness of I&V in improving language models on both concept-to-sentence and concept-to-story generation tasks.

**Summary Of The Review:**

Although the paper is not technically innovative, I do think AMR is a better form of intermediate representation. But my other worry is how to get a good AMR parsing tool, because there is relatively little annotated AMR-text parallel data, and I wonder if the trained AMR tools are ready for use. Overall, this work has a certain contribution. Therefore, I suggest accepting it.

---

> ### Author Response · Authors · 2021-11-18
> **Response to Reviewer 9SQy (Part 1)**
>
> Thank you for your positive review and very helpful comments!  Below are our responses and we have updated the draft accordingly.
>
> >1. The novelty of the whole method may be limited. On the one hand, previous work has used SRL as an intermediate representation, and this work simply replaces it with AMR. On the other hand, pretrained language models are directly adopted in the two stages of the proposed framework.
>
> Our contribution and novelty lie in building the contextualized imagination module, which is not simply replacing SRL with AMR and applying pre-trained LMs.
>
> Note that prior works using SRL as intermediate representations for story generation do not take concepts as constraints. The problem we are tackling in this work is to construct a contextualized graph to connect the given concepts plus implicit concepts, which has not been well explored before.
>
> To tackle this problem, we make the following efforts to turn the graph construction into a sequence-to-sequence problem which allows us to properly utilize a generative LM.
> - (1) We randomize the concepts during training such that the graph construction can be invariant to a specific order.
> - (2) We randomly drop 0~35% of concepts during training such that the imagination module learns to infer the implicit concepts.
> - (3) We adopt the PENMAN format to linearize our SKGs to provide a consistent pattern of the plausible graphs such that the imagination module can effectively learn the mapping.
>
> The adoption of AMR also opens up an opportunity for us to unify a diverse set of scene knowledge from different resources as indirect supervision as detailed in Sec 2.2, while prior works train their intermediate predictors using the task datasets only.
>
>
> >2. The parsing quality of the AMR parsing tool.
>
> We adopt a python package called AMRLib which conducts AMR parsing based on a fine-tuned T5 model. The model achieved ​​81 SMATCH score on LDC2020T02 dataset. We also conduct a quick human evaluation of 50 random AMR graphs extracted from our corpus. Three annotators are asked to score each AMR graph by either 1 (correct parsing) or 0 (incorrect parsing). On average 89.3% of cases are regarded as correct, which indicates a reasonably good parsing quality.
>
> >3. The evaluation of the generated AMR of the first stage was not given in the experiment. Will the generated AMR graph contain a lot of noise?
>
> Since there are no ground truth SKGs annotated in downstream datasets, we use the silver-standard SKGs as reference to give an estimation of the quality of the generated SKGs. We focus on recall since the silver-standard SKGs may not cover all the plausible scenes. Our evaluation considers the following three metrics:
> - (1) Average recall of the given concepts (Explicit Concepts) to examine whether an SKG contains all the given concepts.
> - (2) Average recall of the implicit concepts (Implicit Concepts) to examine whether an SKG also contains implicit concepts. The implicit concepts for reference are the nodes from the silver-standard SKGs excluding the given concepts.
> - (3) Average recall of the relations (Relation) to examine the proportion of the referenced relations that are covered by the generated SKGs. Here, a relation is considered as correct only if the head concept, relation and the tail concept all match the reference.
>
> The results (in percentage) are shown below.
>
> | Metric                   | CommonGen | VIST | ROC  |
> | ----------------------- | -----------------: | ------: | -------: |
> | Explicit Concepts |             99.81 | 99.96 | 99.95 |
> | Implicit Concepts |             17.07 | 35.66 | 61.86 |
> | Relation               |             72.10 | 68.19 | 74.04 |
>
> The results indicate a fairly good quality of the generated SKGs, which connect all the given concepts for over 99% of the cases and have a large overlap (over 68%) with the silver-standard SKGs. Note that the particular low recall on implicit concepts is due to the fact that there can be many different implicit concepts to constitute a complete SKG.
>
> In our submission, we also have a human evaluation study to manually examine the quality of the generated SKGs by 2 aspects, i.e. Completeness and Commonsense. Completeness is to assess whether the generated SKG covers all the concepts including the implicit ones. Commonsense is to assess whether the SKG follows human common sense. After the submission, we updated the human evaluation instruction and invited more annotators (5 in total). The updated results shown below indicate that the generated SKGs are of high quality by human judgment with a fair agreement (Fleiss’ kappa k=0.21).
>
> | Metric               | CommonGen | VIST | ROC  |
> | -------------------- | -----------------: | ------: | -------: |
> | Completeness  |             97.30 | 93.80 | 95.70 |
> | Commonsense |             90.15 | 89.70 | 86.60 |

---

> > ### Author Response · Authors · 2021-11-18
> > **Response to Reviewer 9SQy (Part 2)**
> >
> > >4. Add results from baselines to the human evaluation study.
> >
> > The goal of our human evaluation study is to manually inspect the quality of the predicted SKGs and the resulting text. Other intermediate representations adopted by baselines such as SRL are not directly comparable with SKG since they adopt different edge/relation schemes. We will carefully design a more systematic human study in our final version, to allow comparison with baselines.

---

> > > ### Author Response · Authors · 2021-11-20
> > > **Looking forward to hearing from you**
> > >
> > > Dear Reviewer 9SQy,
> > >
> > > We want to send you a friendly reminder for the discussion. Here is a summary of our response for your valuable feedback!
> > >
> > > - To help you better distinguish our work from prior works, we recapped the major contributions in the general response, and the novelty of this work in the individual response to you.
> > > - We provided both automatic and human evaluation results for the AMR parsing tool used in our work and the SKGs predicted by our imagination module. The results show that both extracted and predicted SKGs are of high quality.
> > >
> > > We thank you again for your valuable comments, and we would appreciate it if you could reconsider the evaluation of our work based on our response. We are also happy to extend our response if you have any other concerns.
> > >
> > > Thanks.

---

### Official Review · Reviewer_6Z6c · 2021-11-04

**Correctness:** 3
**Technical Novelty And Significance:** 3
**Empirical Novelty And Significance:** 3
**Recommendation:** 6
**Confidence:** 4

**Main Review:**

Strengths:

- The core task is well defined and motivated. "What is needed to get models to generate through commonsense reasoning."

- The paper is well written overall and is easy to follow along, model design choices are mostly explained.

- The paper compares to multiple existing methods and the line up of related work baselines strengthens the work.


Clarifications/Concerns/Weaknesses:

- My reading of this is that two of the main contributions are in the iterative imagine and verbalize bits. Two critical ablations that don't show up is are those testing: (1) the two-step process: is the two step process even necessary? How does just crunching both transformers in Figure 3 into one perform? (2) the iterative process (Sec. 2.4): going one step further, its not clear that the iterative process helps by going one sentence at a time. E.g. ROC Stories is relatively small in terms of story length and should allow for giving all 4 sentences as input for context for generation.

- The reasoning for the proposed model underperforming KFCNet (Li et al 2021) is also unconvincing. Are there exact numbers on how much test set concept coverage the KFCNet method has in their prototypes vs. your method?

- In the "Does imagination allow models to learn (faster) with less data?" section, a methodology with multiple random seeds, splits etc etc. I'm unclear if this methodology of multiple random seeds etc is used for the rest of the experiments as well. Further, no mention of statistical significance is made (except in Table 1) - this would be necessary to back up some of the analysis especially given the rather close margin between multiple ablations/experiments.

- Move more human subject studies details to the main paper from Appendix A2 (at the expense of some of the equations such as the autoregressive decoding loss and model implementation details perhaps). n=3 as well as the relatively low kappa's (on ROC Stories esp) makes the human studies portion impossible to draw conclusions from. Adding qualitative examples of the SKGs generated and the actual stories/sentences that are verbalized from those would *also* be necessary to make any meaningful conclusions.

- I generally hate making novelty based arguments, but in this particular case - the method is very similar to many others that have come before, both in storytelling and other forms of generation. The plan with structured representations and then verabalize paradigm is very well know (as seen simply in the author's related work). Given this and the concerns regarding the experiments expressed earlier, I am not sure how this paper's contributions are situated. What is the contribution here? The imagine-and-verbalize paradigm? The way to imagine using SKGs by decoding linearized graphs? The unified schema of the SKGs themselves?

Minor - Appendix A.1 - please add details on how many relations you're crunching into how many others (this is mostly a clarity issue)

**Summary Of The Paper:**

The paper introduces an Imagine and Verbalize two-step method for generative commonsense reasoning. The system first imagines a scene in the form of a linearized SKG and uses that as input for a second model that verbalizes it into (more) human readable text. The method is tested on the Concept2Sentence and Concept2Story tasks and compared to multiple baselines drawn from related works.

**Summary Of The Review:**

Overall I think that the paper is well written and has potential but is not ready to be published in its current stage with all the weaknesses mentioned.


====Rebuttal Update====
As written out in the rebuttal response comment below, I am relatively satisfied by the changes the authors have made and recommend that this paper be accepted.

---

> ### Author Response · Authors · 2021-11-18
> **Response to Reviewer 6Z6c (Part 1)**
>
> Thank you for your helpful comments and insightful questions! Below are our responses and we have also updated the draft accordingly.
>
> >1.  “The method is very similar to many others that have come before, both in storytelling and other forms of generation. The plan with structured representations and then verbalize paradigm is very well known (as seen simply in the author's related work). Given this and the concerns regarding the experiments expressed earlier, I am not sure how this paper's contributions are situated. What is the contribution here? The imagine-and-verbalize paradigm? The way to imagine using SKGs by decoding linearized graphs? The unified schema of the SKGs themselves?”
>
> Thank you for the question. We want to clarify that the imagination-and-verbalization paradigm is not the main contribution as claimed for this work. Rather, our major contribution is to develop a **contextualized imagination module** to facilitate generative CSR (see 4th paragraph of the Introduction). The proposed contextualized scene imagination process 1) tackles a challenging (constrained) graph generation problem (i.e, node-to-graph) which prior works have not considered in the context of text generation, 2) adopts a more unified intermediate knowledge representation (i.e., “SKG”) compared to prior works that consider only discrete or linear narrative plots, which allows us to 3) leverage richer indirect supervision signals while prior works do not, and thus 4) can be a transferable and dataset-agnostic module.
>
> __1) A challenging graph generation problem: contextualized scene imagination__
> - Prior works proposing different intermediate representations for story generation do not consider the concept set as a structural constraint. They only model the mapping from a given context to a story plot either as keywords or predicate-arguments. In our experiments, we adapted their method to take concepts as input to work with our problem setting. The concept-to-SKG modeling is a challenging research problem that is different from what prior works consider in story generation. In particular, this problem requires compositional generalization ability, which is to infer the relations for an unseen concept set and also the implicit concepts that are not given to constitute a complete graph.
>
> __2) A more unified intermediate representation__
> - The plan-and-write model uses keywords as the intermediate representation, which does not represent relations between concepts as a non-linear plot structure. The action-plan model uses SRL (predicates and arguments) plots which do provide relational structure but the nodes are phrases and not guaranteed to cover all the given concepts. We adopt SKG as the intermediate representation since it abstracts away from linguistics more than the SRL plots do and thus the resulting SKGs provide denser supervision on how to organize concepts coherently compared with sparse text compared with target sentences or the less abstractive SRL. In addition, the nodes on the SKG are more conceptualized (compared to phrases in SRL) and thus we can leverage the nodes as pseudo concepts to train a concept-to-SKG predictor (the imagination module). Experimental results shown in Figure 5 validate that SKG is overall a superior representation which leads to better performance on three datasets and the performance gain is larger when less training data is used.
>
> __3) Flexibility to leverage a variety of knowledge sources as indirect supervision__
> - Prior works train their intermediate predictor purely based on the task training data. Our SKG offers a more unified representation scheme and thus opens up an opportunity to unify a diverse set of scene knowledge besides the task training data as indirect supervision. As evidenced by Table 4 in the paper, the collected SKGs together form a more comprehensive distribution of the plausible scenes than the task training data does. This allows our system to generalize better.
>
> We will revise the paper to better stress these key contributions in our final version.

---

> > ### Author Response · Authors · 2021-11-18
> > **Response to Reviewer 6Z6c (Part 2)**
> >
> > __4) A transferable and dataset-agnostic module__
> > We train the imagination module with 2 stages, i.e., continual-pretraining over collected SKGs and fine-tuning over silver-standard SKGs from the task datasets. The second stage is optional and thus our imagination module can be transferable and applied to different datasets. Indeed the imagination module that we apply for all the experiments on CommonGen is not fine-tuned. We fine-tune the imagination module since it can benefit from observing the task distribution of the story context.
> >
> > To demonstrate that a general imagination module can still lead to improvement in the concept-to-story task, we conduct an ablation study on ROC dataset where the imagination module is not fine-tuned with the silver-standard SKGs from the task training set. We then train the verbalization in both full training and low-resource settings. The results (SPICE scores) for ROC dataset below show that using the general imagination leads to a certain performance drop compared with a fine-tuned imagination module but still significantly outperforms the baseline in the low-resource setting (Training size={50,500}) with p-value <0.01.
> >
> > | # Training          | 50       | 500   | 5000  | Full    |
> > | --------------------- | -------: | ------: | -------: | -------: |
> > | Node2Text         | 22.86 | 41.81 | 54.67 | 57.66 |
> > | I&V (general)     | 28.46 | 44.49 | 51.38 | 57.93 |
> > | I&V (fine-tuned) | **33.08** | **49.95** | **57.24** | **60.63** |
> >
> > We also include the result of our system on CommonGen (in-house) below. The table shows that the **general imagination** which does not require fine-tuning on CommonGen yields the best performance of our system and it outperforms all the baselines including KFCNet significantly (p-value < 0.01) in the low-resource setting. When a full training set is used, KFCNet does not outperform our general I&V significantly (p-value > 0.05). You can check Figure 5 for a full comparison.
> > - CommonGen
> >
> > | # Training          | 50       | 500   | 5000  | Full    |
> > | --------------------- | -------: | ------: | -------: | -------: |
> > | Node2Text         | 24.14 | 28.74 | 31.29 | 31.57 |
> > | KFCNet             | 23.13 | 23.26 | 28.39 | **33.52** |
> > | I&V (general)     | **27.52** | **31.37** | **32.68** | 33.49 |
> > | I&V (fine-tuned) | - | - | - | 32.92 |
> >
> > >2.1 Two critical ablations that don't show up are those testing: (1) the two-step process: is the two step process even necessary? How does just crunching both transformers in Figure 3 into one perform?
> >
> > We want to clarify that the two-step process is the core design for our proposed “Imagine-and-Verbalize'' approach (Section 2.1), where (1) the imagination is conditioned on the provided context as well as all the previously generated sentences (which is also why we call our method as contextualized imagination) and (2) the verbalization is conditioned on the scene graph generated by the imagination module. For the concept2story task, at the i-th iteration, the imagination module requires the verbalization module to generate the i-th sentence as part of the input to the imagination module first. Then the imagination can go on to generate the i+1 SKG. With this design, the generated SKG at each step can be more relevant to the current storyline. Meanwhile, the verbalization module can focus on generating one sentence at a time based on an explicit alignment between the i-th SKG and the i-th sentence.
> >
> > >2.2 Two critical ablations that don't show are those testing: (2) the iterative process (Sec. 2.4): going one step further, it's not clear that the iterative process helps by going one sentence at a time. E.g. ROC Stories is relatively small in terms of story length and should allow for giving all 4 sentences as input for context for generation.
> >
> > We have an ablation study included in Appendix A.9 to verify that the contextualization is helpful and the iterative process is more effective than 1) generating all the sentences at once, and 2) generating each sentence independently. For baseline 1), we learn an uncontextualized imagination module which only takes concepts as input and does not need the previously generated context. We apply the uncontextualized imagination module to generate all the SKGs at once and then the verbalization module generates all the target sentences at once by taking the provided context, all the concept sets and all the SKGs as input. For baseline 2), we still use the contextualized imagination module to generate an SKG at a time. But we learn a verbalization module which does not take the previously generated sentences as input and thus generates each target sentence independently. We conduct the ablation study on the two datasets of the concept2story task (we do not consider the CommonGen benchmark here since there is only one target sentence to be generated in CommonGen).

---

> > > ### Author Response · Authors · 2021-11-18
> > > **Response to Reviewer 6Z6c (Part 3)**
> > >
> > > (Cont.) The results of the average SPICE scores from 3 runs are shown below. The two baselines are both outperformed by our iterative approach, which verifies that the previously generated sentences are important for both imagination and verbalization and thus the iterative generation process is necessary.
> > >
> > > | Generation Process | VIST | ROC |
> > > | -------------------------- | ------: | ------: |
> > > | all at once                | 57.16 | 54.83 |
> > > | independent            | 27.01 | 36.42 |
> > > | iterative                   | **59.21** | **60.63** |
> > >
> > > >3. The reasoning for the proposed model underperforming KFCNet (Li et al 2021) is also unconvincing. Are there exact numbers on how much test set concept coverage the KFCNet method has in their prototypes vs. your method?
> > >
> > > Our analysis of the prototypes used by KFCNet reveals that 97.4% of the cases in the official test set have perfectly matched prototypes, i.e., sentences containing all the given concepts. We also conduct an evaluation on a baseline where the prototype containing the most number of concepts is taken directly to be the predicted sentence (without any kind of reasoning involved) and it already achieves 26.62 on SPICE. This demonstrates that it is easier for the prototype-based models to predict the sentences by rephrasing the prototypes which have high coverage of the concepts. In our system, when we train the imagination module, we specifically discard SKGs whose nodes constitute a superset of any concept sets observed in the downstream tasks including CommonGen. Thus, our imagination module relies on compositional generalization to predict the SKG.
> > >
> > > Furthermore, we also analyze the concept coverage of the prototypes retrieved by KFCNet on the two story datasets which can be seen as more “out-of-domain”. We find that the perfect matching rate drops to 26.95% and 29.64% on ROC and VIST datasets, respectively. That explains why our method outperforms the Prototype baseline on these two datasets and reveals a shortcoming of KFCNet, which is to rely on in-domain prototypes to obtain a good performance.
> > >
> > > >4. Is the methodology of multiple random seeds etc used for the rest of the experiments as well?
> > >
> > > Yes. For the rest of the experiments (except the submission to CommonGen leaderboard) which use all the training data, we report the average performance from 3 trials with the same set of random seeds. No randomness is involved in sampling the training data.
> > >
> > > >5. No mention of statistical significance is made (except in Table 1)
> > >
> > > Thanks for bringing this up. We conducted the statistical significance analysis on the experiments involving baselines, which include the ablation study on the backend LM used by the verbalization module (Figure 4) and the low-resource experiment (Figure 5). The p-values are shown below, where < 0.01 indicates a significant improvement and < 0.05 indicates a fairly significant improvement, and NA indicates cases where our method does not outperform the best baseline. We have included this analysis in Appendix A.3.
> > >
> > > - For Figure 4:
> > >
> > > | bart-base | bart-bart | t5-base | t5-large |
> > > | ------------: | -----------: | ---------: | ---------: |
> > > |      < 0.01 |     < 0.01 |   < 0.01 |   < 0.01 |
> > >
> > > - For Figure 5:
> > >
> > > | # Training Examples | CommonGen | VIST  | ROC  |
> > > | --------------------------- | -----------------: | -------: | --------: |
> > > | 50                             |            < 0.01 | < 0.01 | < 0.01 |
> > > | 500                           |            < 0.01 | < 0.01 | < 0.01 |
> > > | 5000                         |            < 0.01 |      NA | < 0.01 |
> > > | All                             |                 NA | < 0.05 | < 0.01 |

---

> > > > ### Author Response · Authors · 2021-11-18
> > > > **Response to Reviewer 6Z6c (Part 4)**
> > > >
> > > > >6. Move more human subject studies details to the main paper from Appendix A2 (at the expense of some of the equations such as the autoregressive decoding loss and model implementation details perhaps). n=3 as well as the relatively low kappa's (on ROC Stories esp) makes the human studies portion impossible to draw conclusions from.
> > > >
> > > > We have updated our human evaluation study and moved the details to Sec. 4.3. To facilitate annotators’ understanding of the evaluation task and AMR language, we provided more detailed instructions and concrete examples of AMR relations in our new round of the study. We also invited more annotators (5 in total) and asked them to evaluate *Completeness* -- whether the generated SKG covers all the concepts (both given and implicit) and *Similarity* -- whether the resulting text is similar to the ground-truth as two additional metrics. To allow more options of annotation, we changed the scoring scale to {0, 0.5, 1.0} (previously it was {0, 1}).
> > > >
> > > > The updated results shown below indicate that the generated SKGs are of high quality by human judgement with a fair agreement (Fleiss’ kappa k=0.21). Note that the updated IAA score is not comparable with the one we reported in the original submission since the number of scoring choices is changed.
> > > >
> > > > | Dataset          | Completeness | CommonSense  | Alignment | Similarity |
> > > > | ------------------ | ------------------: | ---------------------: | ------------: | ------------: |
> > > > | CommonGen | 97.30 | 90.15 | 89.90 | 88.30 |
> > > > | VIST              | 93.80 | 89.70 | 91.40 | 76.20 |
> > > > | ROC              | 95.70 | 86.60 | 87.80 | 75.68|
> > > >
> > > > >7. Adding qualitative examples of the SKGs generated and the actual stories/sentences that are verbalized from those would also be necessary to make any meaningful conclusions.
> > > >
> > > > We summarize how imagination can help improve the commonsense aspect of the text generation below. As a comparison, we also show the results from the Node2Text baseline which does not predict an SKG as the intermediate step. We organize the results based on 5 (not necessarily exclusive) types of errors made by Node2Text, which include incorrect role attribution to 1) agent (*dog* instead of *person* throwing a ball), 2) action (a *stick* of tongue instead of *sticking out* his tongue) or 3) object (wearing *phone* instead of *glasses*), 4) failing to infer the implicit concepts and 5) misunderstanding the relations between events.
> > > >
> > > > We show two examples of the analysis below and you can find more examples in Appendix A.4.
> > > > - Example 1 (incorrect agent)
> > > >
> > > > Input concepts: {ski, rope, hold, **boat**, pull}
> > > >
> > > > Prediction w/o imagination: A woman skis downhill as she pulls a boat holding a rope.
> > > >
> > > > Prediction w/ imagination: A boat pulls a skier who is holding a rope.
> > > >
> > > > Generated SKG: (pull<:ARG0> boat), (pull<:ARG1> person), (ski<:ARG0> person), (hold<:ARG0> person), (hold<:ARG1> rope)
> > > > - Example 2 (incorrect object)
> > > >
> > > > Input concepts: {wear, talk, **phone**}
> > > >
> > > > Prediction w/o imagination: A woman is wearing a cell phone and talking to the camera.
> > > >
> > > > Prediction w/ imagination: A man wearing glasses is talking on the phone.
> > > >
> > > > Generated SKG: (talk<:ARG0> man), (wear<:ARG0> man), (wear<:ARG1> glasses), (talk<:medium> phone)
> > > >
> > > > >8. How many relations you're crunching into how many others.
> > > >
> > > > There are in total 3,858 relation types in our processed VisualGenome dataset due to the noisy annotation. We map these relations into 8 relations, including the 7 relations described in Appendix A.1 and an “other” relation for those not meeting the mapping criteria. The 7 non-”other” relations make up 97.73% of the triplets in VisualGenome. We have added this detail to A.1.

---

> > > > > ### Author Response · Authors · 2021-11-20
> > > > > **Looking forward to hearing from you**
> > > > >
> > > > > Dear Reviewer 6Z6c,
> > > > >
> > > > > We want to send you a friendly reminder for the discussion. Here is a summary of our response for your valuable feedback!
> > > > > - We recapped the major contributions to help you better distinguish our work from prior works.
> > > > > - We demonstrated the necessity of adopting a two-step (imagination and verbalization) paradigm and how imagination helps story generation with an iterative process by ablation studies.
> > > > > - We provided an exact number of concept coverage of the prototypes used in KFCNet.
> > > > > - We clarified that the randomness control is consistent in our experiments and the number of relations mapped to AMR in VisualGenome.
> > > > > - We provided the statistical significance analysis of our experiments and qualitative analysis of the model’s behavior with and without imagination.
> > > > > - We re-conducted our human evaluation with a more rigorous design.
> > > > >
> > > > > We thank you again for your valuable comments, and we would appreciate it if you could reconsider the evaluation of our work based on our response. We are also happy to extend our response if you have any other concerns.
> > > > >
> > > > > Thanks.

---

> > > > > > ### Author Response · Authors · 2021-11-22
> > > > > > **Friendly reminder for discussion**
> > > > > >
> > > > > > Hi Reviewer 6Z6c! The rebuttal period is ending very soon, but we haven't heard your thoughts about our response and updated paper. We are happy to extend our clarification if necessary. And any feedback would be greatly appreciated!

---

### Author Response · Authors · 2021-11-18
**General Response (Part 1)**

We thank all of the reviewers for their valuable comments. Here is a summary of the updates we made in the paper based on the comments and questions:
- Updated human evaluation: 1) Added two additional evaluation metrics, namely Completeness (whether the SKG covers all the concepts) and Similarity (whether the resulting sentences are similar to the ground-truth text). 2) Invited another two human annotators (5 in total). 3) Changed the scoring scale to {0, 0.5, 1.0} to allow more options of annotation.
- Added analysis and ablation studies: 1) Statistical analysis on experiments involving baselines in Appendix A.3; 2) Qualitative analysis on comparing models with/without imagination in Appendix A.4; 3) Quality evaluation of the generated SKG in Appendix A.5; 4) Ablation study on what inputs (context/concepts/SKG) are fed to the imagination and verbalization modules respectively in Appendix A.6; 5) Ablation study on what SKGs (generated or silver-standard) are used to learn the verbalization in Appendix A.7. 6) Ablation study on the design of the generation process in Appendix A.9.
- Added experimental results: 1) Added the results of I&V with a general imagination module which is not fine-tuned over the story datasets in Figure 5. 2) Added the results of the Action-Plan baseline on CommonGen (in-house) in Figure 5.
- Added clarification:  1) Clarified that Node2Text also takes previously generated sentences as input when performing the concept-to-story task in Sec. 3; 2) Provided the exact coverage of testing concept sets of the prototypes used in the KFCNet and the fact that we filtered out overlapped concept sets when collecting SKGs in Sec 4.1; 3) Provided the number of relations from VisualGenome which are mapped to AMR in appendix A.1.
- Reorganized paper sections: 1) Moved the details of human evaluation to Sec. 4.3 in the main paper; 2) Moved implementation details to Appendix A.2.
- Improved the figures: 1) Changed the starting point of the y-axis in Figure 4 to facilitate comparison. 2) Changed the style of Figure 4 and Figure 5 to have a better illustration.
- Fixed typos: 1) pretrainined → pretrained in page 3; 2) moperformance → performance in page 8; 3) Imagine → I&V in Figure 5.

---

> ### Author Response · Authors · 2021-11-18
> **General Response (Part 2)**
>
> We would also like to recap the main contributions and novelty here. We would add a paragraph summarizing our contributions at the end of the introduction in the final version of our paper.
>
> Our major contribution is to build a **contextualized imagination module** which 1) tackles a challenging (constrained) graph generation problem (i.e, node-to-graph) which prior works have not considered in the context of text generation, 2) adopts a more unified intermediate knowledge representation (i.e., “SKG”) compared to prior works that consider only discrete or linear narrative plots, which allows us to 3) leverage richer indirect supervision signals while prior works do not and thus 4) can be a transferable and dataset-agnostic module.
>
> __1. A challenging graph generation problem: contextualized scene imagination__
> - Prior works proposing different intermediate representations for story generation do not consider the concept set as a structural constraint. They only model the mapping from a given context to a story plot either as keywords or predicate-arguments. In our experiments, we adapted their method to take concepts as input as well to work with our problem setting. The concept-to-SKG modeling is a challenging research problem which is different from what prior works consider in story generation. In particular, this problem requires compositional generalization ability, which is to infer the relations for an unseen concept set and also the implicit concepts that are not given to constitute a complete graph.
>
> __2. A more unified intermediate representation__
> - The plan-and-write model uses keywords as the intermediate representation, which does not represent relations between concepts as a non-linear plot structure. The action-plan model uses SRL (predicates and arguments) plots which do provide relational structure but the nodes are phrases and not guaranteed to cover all the given concepts. We adopt SKG as the intermediate representation since it abstracts away from linguistics more than the SRL plots do and thus the resulting SKGs provide denser supervision on how to organize concepts coherently compared with sparse text compared with target sentences or the less abstractive SRL. In addition, the nodes on the SKG are more conceptualized (compared to phrases in SRL) and thus we can leverage the nodes as pseudo concepts to train a concept-to-SKG predictor (the imagination module). Experimental results shown in Figure 5 validate that SKG is overall a superior representation which leads to better performance on three datasets and the performance gain is larger when less training data is used.
>
> __3. Flexibility to leverage a variety of knowledge sources as indirect supervision__
> - Prior works train their intermediate predictor purely based on the task training data. Our SKG offers a more unified representation scheme and thus opens up an opportunity to unify a diverse set of scene knowledge besides the task training data as indirect supervision. As evidenced by Table 4 in the paper, the collected SKGs together form a more comprehensive distribution of the plausible scenes than the task training data does. This allows our system to generalize better.
>
> __4. A transferable and dataset-agnostic module__
> - We train the imagination module with 2 stages, i.e., continual-pretraining over collected SKGs and fine-tuning over silver-standard SKGs from the task datasets. The second stage is optional and thus our imagination module can be transferable and applied to different datasets. Indeed the imagination module that we apply for all the experiments on CommonGen is not fine-tuned. We fine-tune the imagination module since it can benefit from observing the task distribution of the story context. To demonstrate that a general imagination module can still lead to improvement in the concept-to-story task, we conduct an ablation study on ROC dataset where the imagination module is not fine-tuned with the silver-standard SKGs from the task training set. We then train the verbalization in both full training and low-resource settings.
>
> The results (SPICE scores) for ROC dataset below show that using the general imagination leads to a certain performance drop compared with a fine-tuned imagination module but still significantly outperforms the baseline in the low-resource setting (Training size={50,500}) with p-value <0.01.
> - ROC
> | # Training          | 50       | 500   | 5000  | Full    |
> | --------------------- | -------: | ------: | -------: | -------: |
> | Node2Text         | 22.86 | 41.81 | 54.67 | 57.66 |
> | I&V (general)     | 28.46 | 44.49 | 51.38 | 57.93 |
> | I&V (fine-tuned) | **33.08** | **49.95** | **57.24** | **60.63** |

---

> > ### Author Response · Authors · 2021-11-18
> > **General Response (Part 3)**
> >
> > We also include the result of our system on CommonGen (in-house) below. The table shows that the **general imagination** which does not require fine-tuning on CommonGen yields the best performance of our system and it outperforms all the baselines including KFCNet significantly (p-value < 0.01) in the low-resource setting. When a full training set is used, KFCNet does not outperform our general I&V significantly (p-value > 0.05). You can check Figure 5 for a full comparison.
> > - CommonGen
> >
> > | # Training          | 50       | 500   | 5000  | Full    |
> > | --------------------- | -------: | ------: | -------: | -------: |
> > | Node2Text         | 24.14 | 28.74 | 31.29 | 31.57 |
> > | KFCNet             | 23.13 | 23.26 | 28.39 | **33.52** |
> > | I&V (general)     | **27.52** | **31.37** | **32.68** | 33.49 |
> > | I&V (fine-tuned) | - | - | - | 32.92 |

---

### Decision · Program_Chairs · 2022-01-20

**Decision:**

Accept (Poster)

**Comment:**

Strengths:
* Strong results across two benchmarks
* Ablation study demonstrates importance of components
* Provides improvements especially in low resource settings
* Well-written paper

Weaknesses:
* Novelty of the method may be limited as previous works have explored structured outputs as intermediate plans
* Not clear method will extend to other domains as decent AMR parses are required to train the imagination module, which might work well on the datasets used (e.g., RocStories), but wouldn't work in settings with more complex language